# Evaluation of ISSA proactive leading indicators for safety, health and well-being: Application of multi-criteria decision-making methods based on hesitant fuzzy

Mina Bargar[1], Mehdi Jahangiri[2]*, Moslem Alimohammadlou[3], Sanaz Karimpour[1], Mojtaba Kamalinia[4]

**1** Student Research Committee, School of Health, Shiraz University of Medical Sciences, Shiraz, Iran, **2** Department of Occupational Health and Safety Engineering, School of Health, Shiraz University of Medical Sciences, Shiraz, Iran, **3** School of Economics, Department of Economics, Management and Social Sciences Shiraz University, Shiraz, Iran, **4** Department of Occupational Health and Safety Engineering, School of Health, Shiraz University of Medical Sciences, Shiraz, Iran

* Jahangiri_m@sums.ac.ir

## Abstract

### Background

The Vision Zero (VZ) strategy, adopted by the International Social Security Association (ISSA), is based on seven golden rules and 14 safety, health, and well-being (SHW) indicator called VZ Proactive Leading Indicators (VZPLI).

### Methods

The study evaluates the status of SHW VZPLI in Iran's Petrochemical Industries by IAM-VZPLI (ISSA Assessment Method for VZPLI) and proposes a new model for assessing these indicators, called EMA-VZPLI (Extended Method for Assessment of VZPLI). EMA-VZPLI uses an integrated method consisting of multi-criteria decision-making(MCDM), the best-worst method(BWM), Decision-Making Trial and Evaluation Laboratory (DEMATEL), and axiomatic design(AD) based on hesitant fuzzy(HF) sets.

### Results

Weight of indicators and their relationships was determined using hesitant fuzzy best-worst method (HFBWM) and Interval-Valued Hesitant Fuzzy DEMATEL(IVHF-DEMATEL) method respectively. The results showed that the C2 indicator (competent leadership) was the most important, while the C10 indicator (Procurement) was the least important. Additionally, the C2 (competent leadership) and C1 (Visible leadership commitment) indicators were dependent, showing the

**Data availability statement:** All relevant data are within the article and its Supporting Information files.

**Funding:** This study is part of the master's thesis of Ms. Mina Bargar, a master's student at Shiraz University of Medical Sciences. It was financially supported by the Research and Technology Deputy of Shiraz University of Medical Sciences under grant number [Grant Number: 23812]. The funders had no role in study design, data collection and analysis, decision to publish, or preparation of the manuscript.

**Competing interests:** The authors declare that they have no known competing financial interests or personal relationships that could have appeared to influence the work reported in this paper.

**Abbreviation:** VZ, Vision Zero; ISSA, International Social Security Association; SHW, Safety, health, and well-being; VZPLI, Vision Zero Proactive Leading Indicators; SHW VZPLI, Safety, health, and well-being Vision Zero Proactive Leading Indicators; IAM-VZPLI, International Social Security Association Assessment Method for Vision Zero Proactive Leading Indicators; EMA-VZPLI, Extended Method for Assessment of Vision Zero Proactive Leading Indicators; MCDM, multi-criteria decision-making; BWM, best-worst method; DEMATEL, Decision-Making Trial and Evaluation Laboratory; AD, axiomatic design; HF, hesitant fuzzy; HFBWM, hesitant fuzzy best-worst method; IVHF-DEMATEL, Interval-Valued Hesitant Fuzzy Decision-Making Trial and Evaluation Laboratory

highest degree of dependence. The indicators of the 'well-being' aspect in all studied petrochemical companies were worse than those of the safety' and 'health' aspects.

## Conclusion

The results showed a significant difference in ranking of petrochemical companies for VZPLIs in EMA-VZPLI compared to IAM-VZPLI. Therefore, this method could be applied in more accurate assessment of VZPLIs of VZ indicators.

## 1. Introduction

In processing industries, including petrochemicals, accidents involving hazardous chemicals and high temperatures/ pressures occur frequently, resulting in human casualties, economic losses, and environmental degradation [1–3]. Some of the most serious accidents include the Toulouse refinery explosion in 2001 (30 deaths) and the Venezuelan refinery accident in 2012 (39 deaths), highlighting ongoing safety issues in these industries [4]. Petrochemical resources are crucial in the processing industries due to their flammability, processing materials, operating conditions, and accident consequences [5,6]. Occupational health and safety management systems (OHSMS) have been established to prevent workplace accidents and promote occupational health and safety (OHS). These systems manage risk through proactive indicators and active involvement in continuous evaluation of activities[7,8]. In this field, guides and standards such as OHSAS 18001, ILO-OHS2001, AS/NZS4801, PSM, ISO45001, HSE-MS, etc. have been published [5,7]. Despite the widespread implementation of these systems, uncertainty remains about their effectiveness due to the lack of a comprehensive method to evaluate these systems and identify factors that influence their success [8]. Some studies have shown that companies evaluate the effectiveness of the systems based on their own criteria and ignore the functional aspects [9]. The strategy of Vision Zero (VZ) has been successful in overcoming limitations of OHSMS in recent years. Despite its implementation in over 80 countries and involvement of over 10,000 organizations, the VZ strategy has received limited research in industries. The VZ strategy was considered successful in multiple industries despite its short introductory period of about six years [10]. This paper focuses on evaluating a group of proactive leading indicators for safety, health and well-being (SHW) that were published by the International Social Security Association (ISSA). The ISSA introduced the VZ strategy and its fourteen prevention-centered sections at the XXI World Congress on SHW in Singapore, which took place in September 2017 [9]. While the management systems are employed in the field of OHS, they have limitations that support the VZ approach. The VZ strategy differs from traditional management systems in several ways. Traditional systems prioritize risk management, while VZ prioritizes business and leadership. While OHS is prioritized in traditional OHSMS, it is regarded as a value in VZ. Instead of traditional OHS systems that focus on worker behavior and human error, VZ encourages participation and ideation to empower workers and integrate them into the solution. Accidents are seen as

learning opportunities in VZ, while in traditional OHS systems they are seen as signs of failure [2]. The combination of VZ and existing management systems will lead to a more effective outcome. The VZ focuses on the attitude of employees, managers, and workers, not just complying with laws and requirements [2]. In fact, the implementation of OHSMS is one of the VZ's seven golden rules. The fourth rule of VZ, is "ensure a safe and healthy system is well-organized", emphasizing implementation OHSMS. Therefore, VZ has a goal beyond just establishment of an OHSMS and includes all aspects of SHW. Japan's zero accident campaign, founded in 1973 by JISHA, has significantly reduced work-related accidents. The campaign has become a global movement, with major Japanese companies like Toyota, Hyundai, and Panasonic participating. They emphasized that the "New Japan's OHS Systems" is expected to be more practical with the implementation of ISO 45001 to promote a higher level of OHS [10]. Furthermore, Masaho Dohi et al. Studied a Japanese control equipment manufacturing company, focusing on its commitment to safety, ISO 45001 certification, VZ performance, and safety-trained personnel. The study aims to develop safety and security technology, advance production, and develop human resources, with innovation in management systems such as VZ, to advance the Japanese electronics industry [10]. VZ consists of seven golden rules (Table 1) and four principles: life is not negotiable, people make mistakes, handling mental and physical pressure is crucial, and situational prevention is the top priority [11]. "Vision Zero at Work" by Gerard Zwetsloot and Pete Keynes (2022) explores the concept of VZ in the workplace, focusing on zero defects, downtime, and pollution. It emphasizes worker SHW, preventing accidents and injuries. The adoption of VZ is tied to a culture of prevention and is based on the commitment of senior leaders and staff. The study highlights the interconnectedness of the workplace, organization, and national and international policies through VZ implementation [12]. Studies by Zwetsloot Zutslovet et al. (2017) [13] and Er Akan et al. (2022) [14] highlight the importance of commitment, communication, culture, and learning in implementing the zero accident vision (ZAV) in European companies. They suggest identifying risks, conducting risk assessments, examining accidents, implementing safety goals, monitoring progress, correcting errors, and cultivating a preventive culture. The VZ seven golden rules framework was used to develop 14 proactive leading indicators (PLIs) (Table 1), with two indicators for each rule, covering SHW aspects [9]. Three commonly used indicators are results-based, compliance-based, and OHS management process-based. The result-based approach uses lagging indicators, such as the frequency and severity of accidents, which may not accurately measure the quality of existing OHS management and leadership processes [8,9,15].

**Table 1. Vision Zero 7 Golden Rules and 14 proactive leading indicators for SHW [16].**

| Vision zero 7 golden rules | 14 Proactive leading indicators for SHW |
|---|---|
| 1. Take leadership – demonstrate commitment | **C1**-Visible leadership commitment |
| | **C2**- Competent leadership |
| 2. Identify hazards – control risks | **C3**-Evaluating risk management |
| | **C4**-Learning from unplanned events |
| 3. Define targets – develop programs | **C5**-Workplace and job induction |
| | **C6**-Define targets – develop programs |
| 4. Ensure a safe and healthy system – be well-organized | **C7**-Pre-work briefings |
| | **C8**-Planning and organization of work |
| 5. Ensure safety and health in machines, equipment and workplaces | **C9**-Innovation and change |
| | **C10**-Procurement |
| 6. Improve qualifications – develop competence | **C11**-Initial training |
| | **C12**-Refresher training |
| 7. Invest in people – motivate by participation | **C13**-Suggestions for improvement |
| | **C14**-Recognition and reward |

Since no proactive indicators for SHW have been established, businesses focus on preventive measures. Researches show that organizations prioritize health and safety over workplace well-being, while health and risks receive less attention. Social and human values, such as employee trust and physical fitness, are crucial for effective safety management [9]. In the study "Vision Zero: Development of Proactive Leading Indicators (PLIs) for SHW at Work", Gerard Zwetsloot al.'s detail the process of creating these indicators, their measurement and development by companies and countries, and the possibility of creating evaluation case studies to evaluate their effectiveness in real-world settings and identify areas for further development [9]. The VZ SHW PLIs can be utilized in any workplace, business, or industry around the world to prevent accidents and injuries and create a healthy work environment [9,11]. According to the ISSA Assessment Method for Vision Zero Proactive Leading Indicators (IAM-VZPLI) [9], the SHW PLIs can be measured using three different techniques/options of semi-quantitative, quantitative, and qualitative. Qualitative measurement is a checklist approach that focuses on key activities for appropriate SHW processes, benefiting both small and large organizations. It distinguishes between wellbeing, health, and safety aspects and is suitable for companies without SHW experts or specialists [9,16]. The semi-quantitative option measures the frequency of key SHW process activities in a methodical and reliable manner, making it easier for medium-sized businesses to measure and benchmark their performance using a five-point scale ranging from "Always" to" Never. " [9,16]. Quantitative measurement measures the frequency or percentage of key activities and can be applied to internal and external criteria at national and international levels. However, organizations may find it less interesting due to the more work involved in gathering and recording indicator data. These three techniques can be considered part of an evolution, starting with option 1 and progressing to options 2 and 3. Small businesses may prefer the first option, while option two is suitable for most organizations with a limited set of indicators [16].

Larger organizations can use the third option, which allows external benchmarking and distinguishes between five levels of organizational performance: Starting, Learning, Progressing, Advancing, and Achieving. Monthly golden rules results can be derived for proactive use and timely follow-up. Trends can be plotted over the last 12 months, and a guide provides context, background information, and direction on the set of indicators, allowing for prioritization and adapted to different organizational contexts [9,16]. However, IAM-VZPLI, faults to consider experts' hesitancy, the weight and importance of the indicators, and their influence on each other.

The purpose of this research is to overcome these limitations and propose an Extended Method for Assessment of Vision Zero Proactive Leading Indicators (EMA-VZPLI) using the application of Multi-Criteria Decision-Making Methods (MCDMS) including Best-Worst Method (BWM), DEMATEL, and axiomatic design (AD) based on hesitant fuzzy sets (HFS). Using matrices and digraphs, decision-making evaluation is a popular method for deciphering and illustrating the structure of complex systems [17]. This method typically requires addressing significant uncertainties and subjectivities in the decision-making process [17].

In the following, the history and brief explanations about each of these methods and the reasons for choosing them for this study are mentioned.

## 1.1. Best-Worst Method (BWM)

BWM is a novel MCDM technique to determine the weights of indicators, offering fewer pairwise comparisons and more appropriate weighting [18]. Xiaomi Mi et al. (2019) developed three models of BWM for determining criteria priorities using hesitant fuzzy information: The score-based weight-determining model, which uses the score values of hesitant fuzzy elements (HFEs) to indicate the most likely values, the second model, which extends HFEs to equal lengths based on decision-makers' attitudes, and the third model, which computes normalized weights with the highest reliability after going through every possible pairwise comparison value without omitting or adding any information [19]. BWM is an MCDM method which can be used in several phases of solving an MCDM problem. More specifically, it can be used to evaluate the alternatives with respect to the criteria (especially in cases where objective metrics are not available to evaluate the

alternatives). It can also be used to find the importance (weight) of the criteria which are used in finding a solution to satisfy the main goal(s) of the problem [20]. BWM has been used to solve many real-world MCDM problems in areas such as business and economics, health, IT, engineering, education, and agriculture. In principle, wherever the aim is to rank and select an alternative from among a set of alternatives, this method can be used. It can be used by one decision-maker or a group of decision-makers [20].

## 1.2. Dematel

The DEMATEL method was developed by the Geneva Research Center of the Battelle Memorial Institute to solve complex global issues involving structural relationships in complex systems [21]. As an MCDM, DEMATEL can identify patterns of causal relationships between variables, classify complex factors into causes and effects, and give the decision-maker a clear understanding of the relationships that are in place. These capabilities will help to better highlight the positions of factors and the ways in which they interact with one another [17]. Previous studies on MCDM techniques like AHP, ANP, and DEMATEL have overlooked the potential of more sophisticated approaches like BWM in determining weight and ranking indicators. The third limitation is addressed by the proposed approach, which looks into the effects that the indicators give and receive through DEMATEL. Umut Asana et al.'S study "New DEMATEL approach based on hesitant fuzzy sets with interval value" explores the tries the decision making and complex systems analysis using matrices and diagrams.The study proposes a new interval-valued hesitant fuzzy approach for DEMATEL to effectively represent and address uncertainty in expert assessments, highlighting the need for more comprehensive approaches to address uncertainty in decision-making processes [17].

## 1.3. AD based on hesitant fuzzy sets (HFS)

HFS was used to overcome the limitations of current methods like uncertainty, ambiguity, and hesitation. HFS enhances linguistic information, allowing experts to compare options using more expressive language structures like "much more than" and "less than average," thereby reducing doubts and enhancing decision-making [22]. This study uses AD as a method for ranking petrochemical companies. Mustafa Batuhan Ayhan's 2018 study introduces a new supplier selection approach that considers uncertainty in evaluating options and criteria. It introduces hesitant versions of Fuzzy-AHP and Fuzzy Axiomatic Design, allowing decision makers to use ambiguous language and define system and scope design cases in dilemma situations. The study presents a numerical example of six options, assesses decision power, and compares non-hesitant versions of suggested methods [23]. AD is a systematic approach to system design that addresses uncertainty and is widely accepted by researchers and academics. It is distinguished by its independence and informational foundations, and is considered more effective than other methods. AD, first introduced by Suh in 1991, describes design as the dynamic interaction between desired results and effective means [23].

A review of various studies conducted in the field of evaluating proactive indicators in the world shows that researchers have mainly studied the VZ strategy and the importance of its implementation in industries, the importance of commitment, communication, culture and learning for implementing VZ in accidents, misleading thoughts in understanding VZ, using an effective method to improve occupational safety and minimize occupational injuries, OHSMS, efforts to ensure the safety of machinery and human resource development, and renewing the VZ strategy by focusing on how to intervene using a human factors perspective [12–14,24–29]. Other researchers have also introduced proactive indicators for SHW, applied tools for SHW management, and found solutions to complex challenges in the field of SHW globalization [9,30–32]. This measurement models and methods are simple and checklist-based, and no study has used developed multi-criteria decision-making methods and further research seems to be needed to determine the importance, weight, and relationships between these indicators.

Based on the investigations conducted, the distinction and innovation of this study compared to other studies can be noted in the following:

- Generalizing the calculation of the vision zero indicators to the fuzzy space to overcome the uncertainty in the respondents' statements

- Using hesitant fuzzy in calculating the vision zero indicators for better modeling of uncertainties

- Developing the vision zero model by weighting the indicators and examining the effectiveness and impact of the indicators, respectively, using the developed HFBWM method and the developed IVHF-DEMATEL method with the aim of organizing the indicators and preparing a graphic image in order to use it in determining improvement strategies

- Developing the vision zero model scoring method using the developed Hesitant Fuzzy Axiomatic(HF-AD) Design method and applying it for the first time.

This study addresses uncertainty and ranks indicators using AD, BWM, and DEMATEL, using standard fuzzy sets or clean data to address ambiguity. It identifies theoretical gaps in the literature and proposes an integrated MCDM approach based on HSFs to assess ZV proactive leading indicators for SHW. This study is the first to use Interval-Valued Hesitant Fuzzy (IVHFS) together with three other powerful decision-making methods (BWM, DEMATEL, and AD) to deal with uncertainty in the evaluation of VZ proactive leading indicators. This approach can capture both personal uncertainty (an expert's uncertainty about the evaluation) and interpersonal uncertainty (ambiguities in experts' evaluations). Assigning membership degrees to a set of decision-making problems, especially group-based problems, is a difficult task. Other fuzzy-theoretic approaches usually show a moderate level of capability in dealing with this issue. Unlike approaches in the literature, the proposed method does not require additional information that may go beyond the decision-maker's knowledge and skills in managing uncertainty (e.g., data distributions, belief structures, reference series of membership functions, label indices). When decision-makers have difficulty expressing membership degrees in terms of clear numbers, they can simply use possible membership degrees in terms of interval values. This strategy simplifies the experts' judgment. This developed method enables modeling a higher degree of uncertainty compared to what conventional fuzzy sets offer [33]. It replaces the causal diagram in DEMATEL with an influence dependency diagram and strongly integrates IVHFS and DEMATEL without reducing accuracy. The proposed method here uses hesitant fuzzy operations to perform all calculations and does not require converting values to explicit values. This option enables the analysis to store the uncertainties in the cumulative effects of interacting factors while avoiding incorrect decisions due to information loss.

## 2. Material and methods

This descriptive cross-sectional study conducted on 16 petrochemical industries (P1 to P16) in Iran between September 12, 2021 and March 13, 2023. 47 experts in the field of OSH participated in this study. The study participants were selected purposefully based on their knowledge, experience, and expertise in the subject matter. From each petrochemical company, at least 2 experts with at least 5 years of work experience related to OHS education were selected to send the link to the checklists through Google Form for completion by them. The study participants verbally expressed their consent and were initially given explanations about the purpose of the study and the principle of confidentiality of information. Fig 1 shows the phases and steps of the study. This study consists of two phases and seven steps, In the first phase, the weight of the indicators was determined through HF-BWM. The relationships between indicators were determined through the IVHF-DEMATEL method, and also assessing and ranking of petrochemical companies was done through HF-AD. In the second phase evaluation of IAM-VZPL (ISSA guidelines and information sheets) was done and the evaluation results of IAM-VZPL were compared with EMA-VZPLI. The method of doing each phase is explained in detail.

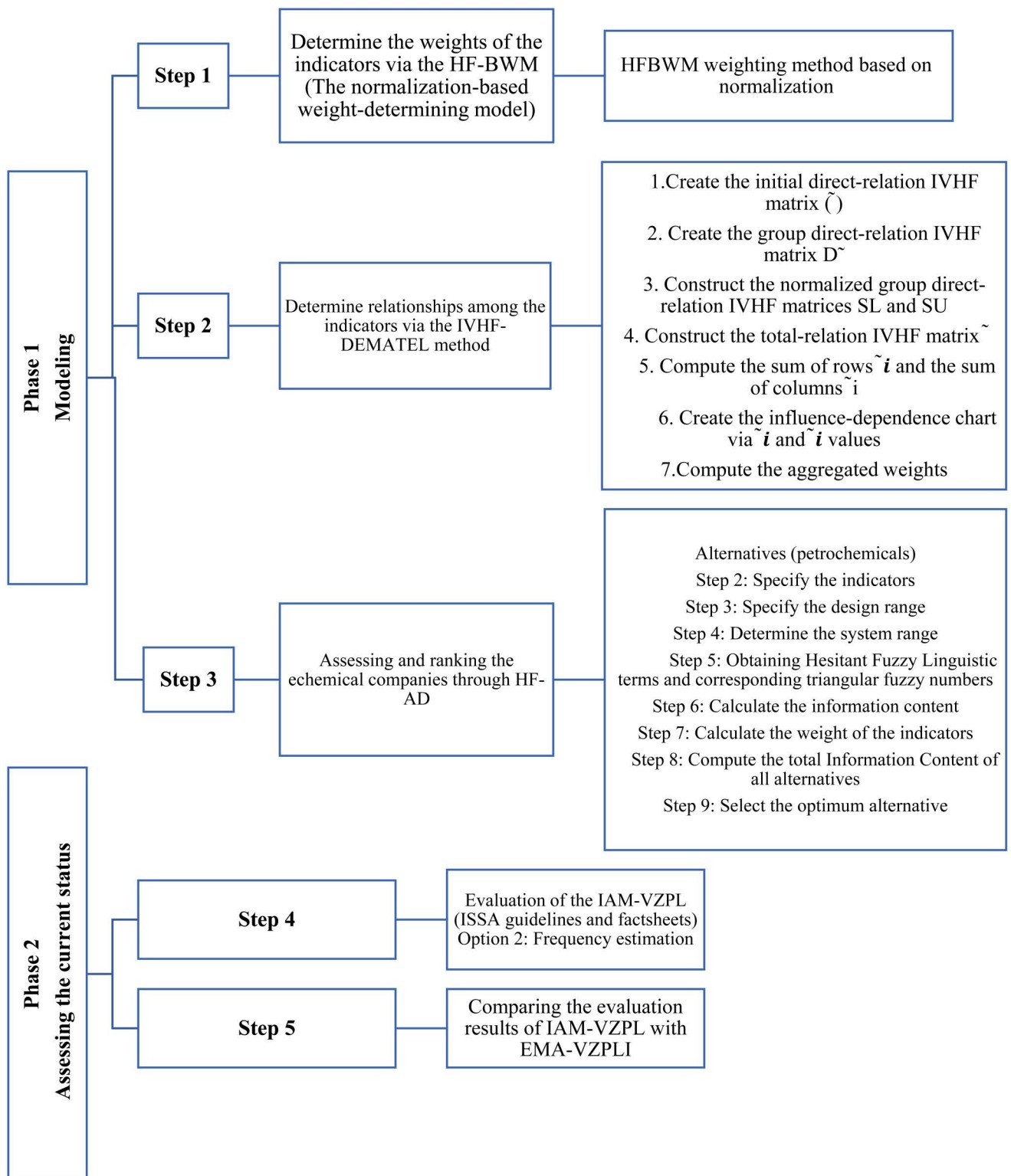

**Fig 1. Phases and Steps of the study.**

## 2.1. Phase 1: modeling

### 2.1.1. Step1: Determine the weights of the PLIs.

The HF-BWM method was used to determine the weights of indicators in a fuzzy, uncertain environment, addressing the subjective nature of MCDM problems. Research indicates that this method can achieve better results than the BWM due to its more consistent comparisons, and it clarifies procedural information. The second model is based on equal-length, extended hesitant fuzzy evaluations that consider decision makers' (DMs') attitudes. This method ensures that no data provided by the DMs is ignored. The hesitant fuzzy weights of indicators with lengths equal to the hesitant fuzzy evaluations can be obtained with this model. Furthermore, these hesitant fuzzy weights obtain more data than crisp weights. We will consider the length as l, assuming a normalized hesitant fuzzy preference value. Let the hesitant fuzzy weight of criterion cj be ωj = (ωj(1), ωj(2),…, ωj (l)), for j = 1, 2,…, n. The hesitant fuzzy weights and their normalized evaluations must meet this condition based on the multiplicative consistency principle [19]:

$$\frac{\omega_B^{(1)}}{\omega_B^{(1)} + \omega_B^{(1)}} = h_{Bj}^{(1)} \, , \; \frac{\omega_B^{(2)}}{\omega_B^{(2)} + \omega_B^{(2)}} = h_{Bj}^{(2)}, \ldots, \; \frac{\omega_B^{(l)}}{\omega_B^{(l)} + \omega_j^{(l)}} = h_{Bj}^{(l)} \tag{1}$$

In order to quantify the degree of coherence displayed by normalized hesitant fuzzy preference vectors, a set of l slack variables can be introduced. The following equation is an illustration of this concept [19]:

$$\left| \omega_B^{(1)} - \left( \omega_B^{(1)} + \omega_j^{(1)} \right) \times \left( h_{Bj}^{(1)} \right) \right| \leq \psi^{(1)}, \left| \omega_B^{(2)} - \left( \omega_B^{(2)} + \omega_j^{(2)} \right) \times \left( h_{Bj}^{(1)} \right) \right| \leq \psi^{(2)}, \ldots, \text{ and } \left| \omega_B^{(l)} - \left( \omega_W^{(l)} + \omega_j^{(l)} \right) \times \left( h_{jW}^{(1)} \right) \right| \leq \psi^{(1)} \tag{2}$$

It's crucial to find a way to normalize the hesitant fuzzy weights. Conventional crisp weights follow the restriction that the total of the weights must equal one. This restriction, however, does not apply to interval-valued weights. A definition of normalized interval-valued weights was inspired by the definition by Sugihara et al. (2004), the constraints of normalized hesitant fuzzy weights are as bellow [19]:

$$j = \left( \omega j \left( 1 \right), \; \omega j \left( 2 \right), \quad \omega j \left( l \right) \right), \, j = 1, 2, .., \, n. \; \omega \tag{3}$$

$$\sum_{j=1}^{n} \omega_j^{\sigma \, (l)} + \omega_t^{\sigma \, (1)} \geq 1 + \; \omega_t^{\sigma \, (l)}, \; \sum_{j=1}^{n} \omega_j^{\sigma \, (1)} + \omega_t^{\sigma \, (l)} \leq 1 + \omega_t^{\sigma \, (1)}, 1 = 1, 2, \ldots, \#\omega \, , \; t = 1, 2, \ldots, \, n$$

The lower and upper bounds of the normalized hesitant fuzzy weight are denoted by ωj (1) and ωj (l), respectively. ω represents the total number of possible values that fall inside ω [19]. The expansion of weight summation in equation (3) aims to eliminate duplication in the solution space of hesitant fuzzy weights. The constraint of the sum of weights equaling one cannot be satisfied within this superfluous solution space. A weight-determining model, Model 2, is developed using normalization principles [19]:

Model 2:

$$Min\psi_R$$

$$s.t = \begin{cases} \left| \omega_B^{(l)} - \left( \omega_B^{(l)} + \omega_j^{(l)} \right) \times \left( h_{Bj}^{(l)} \right) \right| \leq \psi_R \\ \left| \omega_j^{(1)} - \left( \omega_W^{(l)} + \omega_j^{(l)} \right) \times \left( h_{jW}^{(1)} \right) \right| \leq \psi_R \\ \sum_{j=1}^{n} \omega_j^{\sigma(1)} + \omega_t^{\sigma(1)} \leq 1 + \omega_t^{\sigma \, (1)} \\ \sum_{j=1}^{n} \omega_j^{\sigma(1)} + \omega_t^{\sigma(1)} \geq 1 + \omega_t^{\sigma \, (1)} \\ \omega_j \geq 0, \, j = 1, 2, \ldots, n \, l = 1, 2, \ldots, \omega \end{cases}$$

The process of obtaining the normalized hesitant fuzzy weights of indicators can be accomplished by resolving Model 2. Hesitant fuzzy weights offer a more comprehensive amount of information compared to crisp weights. However, additional information is integrated during the normalization process, thereby improving the original assessments [19].

**2.1.2. Step2: Determine relationships between the indicators via the IVHF-DEMATEL.** This step involves evaluating the relationships between the indicators and determining the degree of influence among them using the IVHF-DEMATEL method. The IVHFS approach improves on the traditional DEMATEL technique by considering the uncertainty resulting from human uncertainty. Variations in experts' assessments can be recorded by an IVHFS, and it can preserve data that other methodologies often miss. The following is an outline of the IVHF-DEMATEL approach's steps [17].

**2.1.2.1. Create the initial direct-relation IVHF matrix ($\widetilde{H}$)** Initially, a set of k experts (where k is a member of the set {1,.., k} determine if the variables are related in any way. Professional opinions are converted into In (IVF) numerical values according to instructions of Table 2 [17].

The relationships between the factors F={Fi Ii = 1,2,..., n} are represented by the interval-valued hesitant fuzzy relation matrix (IVHFRM). The following is how these relationships for expert k are expressed:

F={$F_1$ I$_i$ = 1,2,…,n}

$$H^{\sim K} = \begin{bmatrix} 0^\sim & K_{\widetilde{12}} & \cdots & K_{\widetilde{1n}} \\ K_{\widetilde{21}} & 0^\sim & & K_{\widetilde{2n}} \\ \vdots & & \ddots & \vdots \\ K_{\widetilde{n1}} & K_{\widetilde{n2}} & \cdots & 0^\sim \end{bmatrix}$$

(4)

IVHFEs are represented in the matrix as $\widetilde{h}ijk = \{\widetilde{\gamma}ijL, \widetilde{\gamma}\}$, which denotes the possible impact's extent. Within the matrix, i denotes the rows, j corresponds to the columns, and k represents the experts.

**2.1.2.2. Create the group direct-relation IVHF matrix $\widetilde{D}$** The experts' determined level of membership is aggregated into an IVHFE using the IVHFWA operator:

$$d_{\widetilde{ij}} = \oplus_{k=1}^{p}(w_k h_{ij}^{\sim k}) = \left\{ \left[ 1 - \prod_{k=1}^{k}(1-(\gamma_{\widetilde{ij}}^{\sim k})^L)^{w_k}, \ 1 - \prod_{k=1}^{k}\left(1-(\gamma_{\widetilde{ij}}^{\sim k})^U\right)^{w_k} \right] \ \Big| \gamma_{\widetilde{ij}}^{\sim 1} \in h_{\widetilde{ij}}^{\sim 1}, \cdots, \gamma_{\widetilde{ij}}^{\sim k} \in h_{\widetilde{ij}}^{\sim k} \right.$$

(5)

The IVHFE (K) for the decision maker (K) has lower and upper bounds represented by the symbols ($\widetilde{\gamma}ijk$) and ($\widetilde{\gamma}ijk$). It follows that the matrix $\widetilde{D}$, which is the direct-relation matrix's equivalent, is expressed as follows:

$$D^\sim = \begin{bmatrix} 0^\sim & d_{\widetilde{12}} & \cdots & d_{\widetilde{1n}} \\ d_{\widetilde{21}} & 0^\sim & & d_{\widetilde{2n}} \\ \vdots & & \ddots & \vdots \\ d_{\widetilde{n1}} & d_{\widetilde{n2}} & \cdots & 0^\sim \end{bmatrix}$$

(6)

**Table 2. Converting linguistic terms into IVF membership degrees.**

| Linguistic Terms | A | Corresponding interval valued fuzzy membership degrees | |
|---|---|---|---|
| | | L | U |
| No influence (NO) | 0 | 0 | 0 |
| Very low influence (VL) | 1 | 0.25 | 0.35 |
| Low influence (L) | 2 | 0.45 | 0.55 |
| High influence (H) | 3 | 0.65 | 0.75 |
| Very high influence (VH) | 4 | 0.85 | 0.95 |

**2.1.2.3. Construct the normalized group direct-relation IVHF matrices S^L and S^U** At this stage, the main goal is to normalize the matrix. To accomplish this, divide the endpoints of matrix D by the high value of each row sum (d), which is represented as $\widetilde{d_{ij}} = [\widetilde{d_{ij}L}, \widetilde{d_{ij}U}]$. The normalized matrix $\widetilde{S}$ is then produced [17].

$$d = \max_{1 \leq i \leq n} \left\{ \sum_{j=1}^{n} score \; (\widetilde{d_{ij}}^{U}) \right\}$$

(7)

$$S_{ij} = \left\{ \left[ \widetilde{s}_{ij}^{L} \quad \widetilde{s}_{ij}^{U} \right] \right\} = \left\{ \left[ \frac{\widetilde{d_{ij}}^{L}}{d} \quad \frac{\widetilde{d_{ij}}^{U}}{d} \right] \right\}$$

So, the matrix $\widetilde{S}$ splits into two separate hesitant fuzzy matrices, each with an upper and lower limit:

$$s^{L} = \begin{bmatrix} 0^{\sim} & s_{12}^{\sim L} & \cdots & s_{1n}^{\sim L} \\ s_{21}^{\sim L} & 0^{\sim} & & s_{2n}^{\sim L} \\ \vdots & & \ddots & \vdots \\ s_{n1}^{\sim L} & s_{n2}^{\sim L} & \cdots & 0^{\sim} \end{bmatrix}$$

(8)

$$S^{U} = \begin{bmatrix} 0^{\sim} & s_{12}^{\sim U} & \cdots & s_{1n}^{\sim U} \\ s_{21}^{\sim U} & 0^{\sim} & & s_{2n}^{\sim U} \\ \vdots & & \ddots & \vdots \\ s_{n1}^{\sim U} & s_{n2}^{\sim U} & \cdots & 0^{\sim} \end{bmatrix}$$

**2.1.2.4. Construct the total-relation IVHF matrix $\widetilde{T}$** The total direct and indirect relationships between every pair of indicators and the IVHFEs are represented by the matrix $\widetilde{T}$. The formula that follows can be used to calculate matrix $\widetilde{T}$ [17]:

$$T = S \bigoplus S^{2} \oplus \cdots \bigoplus S^{m}$$

(9)

There should be a high enough value for the parameter m. Each lower and upper element of the matrix $\widetilde{S}$ requires individual exponentiation through multiplication and summation operations because hesitant fuzzy sets (HFSs) differ from conventional matrices in the way that multiplication and summation operations work. The following formulas can be used to create the total-relation hesitant fuzzy matrices T L and T U, which represent the upper and lower limits of $\widetilde{T}$:

$$T^{L} = S^{L} \oplus \left( S^{L} \right)^{2} \oplus \cdots \bigoplus \left( S^{L} \right)^{m}$$

(10)

$$T^{U} = S^{U} \oplus \left( S^{U} \right)^{2} \oplus \cdots \bigoplus \left( S^{U} \right)^{m}$$

As a result, the matrices T L and T U are combined in the following way to determine the limit matrix $\widetilde{T}$ [17]:

$$T = \begin{bmatrix} \begin{bmatrix} t_{11}^{L} & t_{11}^{U} \\ t_{21}^{L} & t_{21}^{U} \end{bmatrix} & \begin{bmatrix} t_{12}^{L} & t_{12}^{U} \\ t_{22}^{L} & t_{22}^{U} \end{bmatrix} & \cdots & \begin{bmatrix} t_{1n}^{L} & t_{1n}^{U} \\ t_{2n}^{L} & t_{2n}^{U} \end{bmatrix} \\ \vdots & \vdots & \ddots & \vdots \\ \begin{bmatrix} t_{n1}^{L} & t_{n1}^{U} \end{bmatrix} & \begin{bmatrix} t_{n2}^{L} & t_{n2}^{U} \end{bmatrix} & \cdots & \begin{bmatrix} t_{nn}^{L} & t_{nn}^{U} \end{bmatrix} \end{bmatrix}$$

(11)

The following equation makes it easier to find $S_m$ by calculating the mth power of the $S_L$ and $S_U$ matrices.

$$Sm = \begin{bmatrix} s_{11}^{(m)} & s_{12}^{(m)} & \cdots & s_{1n}^{(m)} \\ s_{21}^{(m)} & s_{22}^{(m)} & \cdots & s_{2n}^{(m)} \\ \vdots & \vdots & \ddots & \vdots \\ s_{n1}^{(m)} & s_{n2}^{(m)} & \cdots & s_{nn}^{(m)} \end{bmatrix}$$

(12)

$$s_{ij}^{(m)} = \left\{ \left[ \left( s_{ij}^{(m)} \right)^L, \left( s_{ij}^{(m)} \right)^U \right] \right\}$$

**2.1.2.5. Compute the sum of rows $\widetilde{r}_i$ and the sum of columns $\widetilde{c}_i$** Calculating the sum of each row and column in an $n \times n$ matrix requires repeating the IVHF sum operator (n-1) times. The sum of the rows, or the cumulative impact of criterion I on the other criteria, is represented by the variable $r\,i$. On the other hand, the total of the columns, denoted by $c\,i$, shows the total influence that criteria I have received from other criteria [17].

$$r = \begin{matrix} \{[r_1^L & r_1^U]\} \\ \{[r_2^L & r_2^U]\} \\ \vdots \\ [\{r_n^L & r_n^U]\} \end{matrix} \qquad c = \begin{matrix} \{[c_1^L & rc_1^U]\} \\ \{[c_2^L & c_2^U]\} \\ \vdots \\ [\{c_n^L & c_n^U]\} \end{matrix}$$

(13)

**2.1.2.6. Create the influence-dependence chart via $\widetilde{r}_i$ and $\widetilde{c}_i$ values** The causal diagram used by the traditional DEMATEL method is built using prominence (Ri+Ci) and relation (Ri-Ci) values, which are usually negative. Since ri and ci values are in line with prominence and relation values, it is recommended to use them in an HFS rather than negative values. Fig 2's influence-dependence chart presents a two-dimensional representation of the data, with the vertical axis representing the total of the columns ($c\,i$) and the horizontal axis representing the sum of the rows ($U\,i$). Four distinct areas are shown in this chart, each of which stands for a factor that is influential, crucial, dependent, or excluded. Any element can be placed anywhere on the chart, and its position will determine which direction it is controlled in. There is an innate connection between the causal diagram and the influence-dependence chart as seen in Fig 2. The causal diagram and influence-dependence chart are interconnected, with areas compatible with (Ri+Ci) and (Ri-Ci) values. Factors in the dependent region have negative values, while those in the influential region have positive values. High (Ri+Ci) values in the critical area indicate high (Ri+Ci) values. The equation determines the boundaries of the four regions by calculating the average of the rows and columns [17].

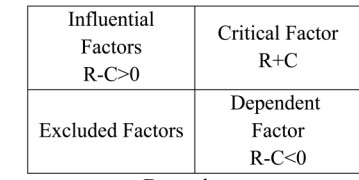

**Fig 2. The influence-dependence chart [17].**

**2.1.2.7. Compute the aggregated weights** The weights obtained from HFBWM and IVHF-DEMATEL are combined, considering the importance of each indicator and their relations and impacts. DEMATEL values are transformed into weights using Equation (14), and Equation (15) is applied to combine these weights with BWM-processed ones.

$$W_{Rj} = \frac{\sqrt{(r_j + c_j)^2 + (r_j - c_j)^2}}{\sum \sqrt{(r_j + c_j)^2 + (r_j - c_j)^2}} \tag{14}$$

$$W_{Fj} = \frac{W_{Rj} \times W_j}{\sum W_{Rj} \times W_j} \tag{15}$$

Where *WRj* demonstrates the weights derived from the DEMATEL approach, *Wj* signifies the weights obtained from the BWM methodology, and *WFj* highlights the ultimate weights.

**2.1.3. Step3: Assessing and ranking the petrochemical companies through HF-AD.** AD was introduced by Suh in 1990 and has been widely used in various engineering fields to improve design-related activities. AD emphasizes logical-cognitive processes and theoretical underpinnings, and can be effectively used in MCDM to find suitable alternatives. The traditional AD methodology's main principles are knowledge and independence, and many research projects have utilized the knowledge principle to solve decision-making-related problems [23]. The principle of independence suggests that the best configuration with minimal knowledge content is chosen from a set of alternatives that meet the principle's requirements [23]. The information axiom, represented by IC, focuses on the probability of meeting a plan's goals. The designer determines the likelihood of achievement based on the design range (DR) and system range (SR) based on individual needs. Observing the common area between these ranges leads to an operational resolution [23] (refer to Fig 3).

To deal with Uncertainty and doubts with expert opinions, the HF-AD method was applied in the current study. Here is an explanation of how the HF-AD method chooses the best alternative [23].

Step 1: A total of 16 production companies was identified as the study alternatives, or the number of petrochemical company.

Step 2: VZ 14 proactive leading indicators for SHW were identified as the indicators for evaluating the alternatives.

Step 3: For each indicator that the designer wanted to achieve, the Design Range DR (desire status) was determined.

Step 4: Using the questionnaire, the System Range SR (current situation) of each alternative was ascertained based on each indicator in which the alternative is capable.

Step 5: Hesitant Fuzzy Linguistic terms and corresponding triangular fuzzy numbers for intangible criteria were defined for each indicator.

Table 3 explains how linguistic terms are translated into Triangular Fuzzy Numbers (TFNs).

Step 6: Determine the Information Content of every alternative for each indicator.

The information content of each criterion i with respect to each alternative k (Iik) can be determined using Eq (16):

$$I\ ik = \log\left(\frac{1}{P\ ik}\right) \tag{16}$$

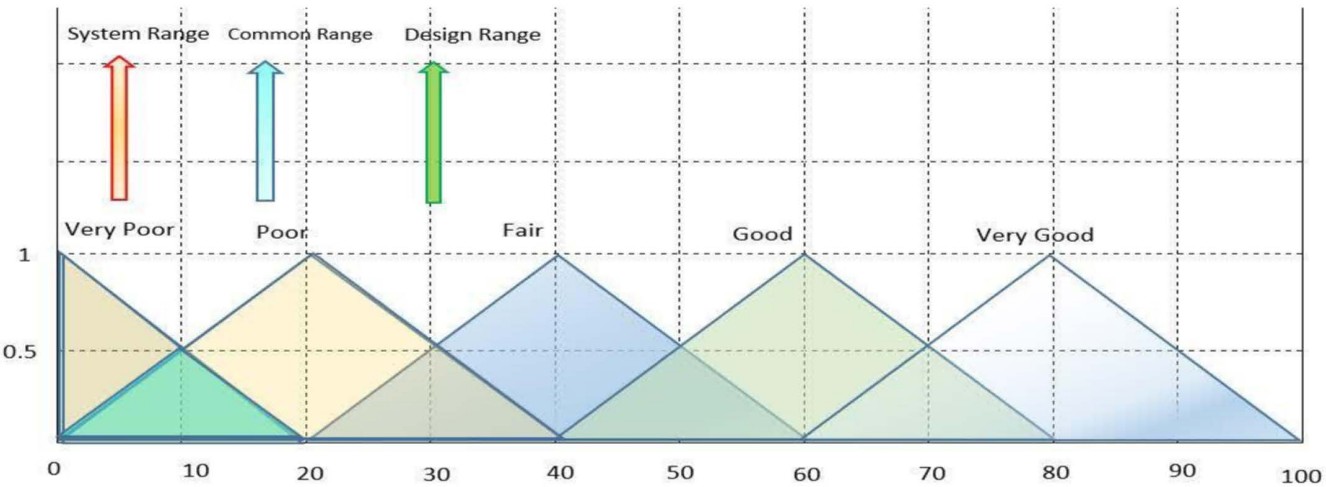

**Fig 3. The intersection area between the SR and the DR.**

Where log represents the base 2 logarithm and *Pik* demonstrates the probability that the alternative k would comply with indicators I [17]. For each alternative, the possibility that the designer describes what he or she hopes to achieve (DR) and what the alternative can do (SR) determines whether or not the criteria are met. To find the preferred solution, one needs to focus on the area where the DR and SR intersect, keeping these two measurements in mind. Therefore, *Pik* can be written as follows in Eq (17):

$$P\,i\,k = \left( \frac{System\ Range\ \cap\ Design\ Range}{System\ Range} \right) = \left( \frac{Common\ Range}{System\ Range} \right) \tag{17}$$

Fig 4 Shows the design range, the system range, and the intersection region.

The information content of criterion i regarding alternative k (*Iik*) can be clarified as follows in light of the previously mentioned statements:

$$Iik = \log 2 \left( \frac{Common\ Range}{System\ Range} \right) \tag{18}$$

Step 7: The study calculates indicator weights using HF BWM and IVHF DEMATEL, integrating the weights to obtain the combined weights. The weights used in this study are obtained by combining the HF-DEMATEL and HF-BWM methods (Eqs (14), (15)).

**Table 3. Converting linguistic terms to triangular fuzzy numbers.**

| Linguistic terms | Triangular fuzzy numbers |
|---|---|
| Very poor | (0%, 0%, 20%) |
| Poor | (0%, 20%, 40%) |
| Fair | (20%, 40%, 60%) |
| Good | (40%, 60%, 80%) |
| Very good | (60%, 80%, 100%) |
| Excellent | (80%, 100%, 100%) |

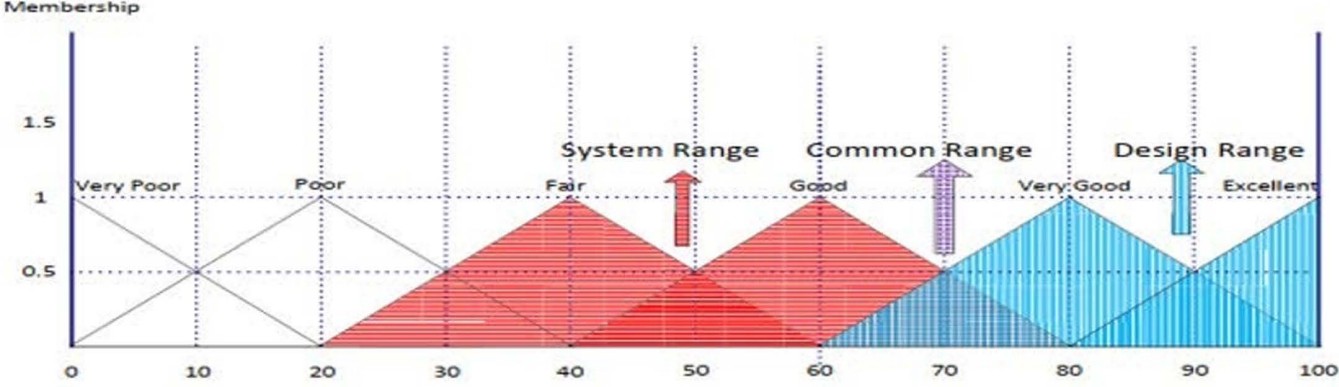

**Fig 4. The design range, the system range, and the intersection region.**

Step 8: Compute the total Information Content of all alternatives. A problem may encompass n distinct criteria. Due to this, the total information content for alternative k equals the sum of information contents of all criteria, as computed through Eq (19):

$$Ik = \sum_{i=1}^{\infty} I\,ik$$

(19)

Step 9: Select the optimum alternative which possesses the least total Information Content. In the final step, the AD technique demonstrates that the alternative with the smallest *Ik* value is the best and the most significant alternative.

### 2.2. Phase 2: Assessing the current status

#### 2.2.1. Step 4: Evaluation of the IAM-VZPL.

The EMA-VZPLI model was used to assess the current and desired status of petrochemical industries in Iran. The IAM-VZPLI method, based on the frequency and extent of implementing performing key activities, was used to assess the present state of VZ -PLIs for SHW. The ISSA PLI for SHW checklist and scoring method were provided by experts from 16 Iran petrochemical companies, each with five years of work experience and professional qualifications in OHS, with an average of three experts per company. A total of 47 experts (an average of two from each company) participated in the study. The following was the scoring system for the frequency of SHW indicators: always or almost always=4, frequently=3, occasionally=2, rarely=1, and never or very rarely=0. The results were graded in 5 levels according to ISSA guidelines (Table 4).

**Table 4. Grading of SHW indicators based on frequency of activities [16].**

| Achieving | 100-81% |
| --- | --- |
| Advancing | 80-61% |
| Progressing | 60-41% |
| Learning | 40-21% |
| Starting | 20-0% |

**2.2.2. Step5: Comparing the evaluation results of IAM-VZPL with EMA-VZPLI.** This section ranks petrochemical companies and compares their performance in proactive leading indicators for SHW using distinguishability index from EMA-VZPLI and IAM-VZPLI, evaluating their key performance. The solutions were presented using proactive leading indicators for SHW and the seven golden rules of Vision Zero to achieve the desired status.

## 2.3. Tools and data collection process

Assessing the current and desired status of the VZ proactive leading indicators for SHW in the study petrochemical companies, an evaluation checklist (16) and a HF-AD questionnaire were prepared electronically (Google Form) for at least two experts (with at least 5 years of relevant work experience and relevant education in OHS. Ten university experts completed the IVHF DEMATEL and HFBWM questionnaires to evaluate the relationships between SHW indicators, their importance, and weight.

# 3. Results

## 3.1. Results of HFBWM

The results of HFBWM showed that, C2 was the most important indicator and C10 was the least important. The reference comparison made by decision makers between the best (CB) and worst (Cw) indicators and the rest of the indicators is as follows:

$$(HF - BO) = (\{0.6, 0.7, 0.55\} \{0.5\} \{0.6, 0.7, 0.8\} \{0.8, 0.7\} \{0.7, 0.9\}$$
$$\{0.7, 0.6\} \{0.7, 0.8, 0.9\} \{0.6, 0.7\} \{0.8, 0.6\} \{0.9\} \{0.7, 0.9, 0.6\}$$
$$\{0.7, 0.9\} \{0.9, 0.7\} \{0.8, 0.7\})$$

$$(HF - OW) = (\{0.9\}\{0.9, 0.85\}\{0.8, 0.75\}\{0.7, 0.75\}\{0.8, 0.6\}\{0.7\}\{0.8, 0.7\}\{0.9\}\{0.75\}$$
$$\{0.9, 0.6\}\{0.5\}\{0.8, 0.6\}\{0.8, 0.7\}\{0.8, 0.55\}\{0.8, 0.65\})$$

In this model, the normalized fuzzy preference vectors were first calculated. Where the normalization parameter is set to zero. In other words, the pessimistic attitude of the experts was considered. Subsequently, the normalized hesitant fuzzy preference vectors were acquired through the following procedure:

$$(HF - BO) = (\{0.55, 0.6, 0.7, 0.55\} \{0.5, 0.5, 0.5, 0.5\} \{0.6, 0.7, 0.8, 0.6\}$$

$$\{0.7, 0.7, 0.7, 0.8\}\{0.7, 0.9, 0.7, 0.7\} \{0.7, 0.6, 0.6, 0.6\} \{0.7, 0.7, 0.8, 0.9\}$$

$$\{0.6, 0.7, 0.6, 0.6\}\{0.8, 0.6, 0.6, 0.6\}\{0.9, 0.9, 0.9, 0.9\}\{0.6, 0.7, 0.9, 0.6\}$$

$$\{0.7, 0.7, 0.7, 0.9\}\{0.9, 0.7, 0.7, 0.7\}\{0.8, 0.7, 0.7, 0.7\}$$

$$(HF - OW) = (\{0.7, 0.7, 0.8, 0.9\} \{0.7, 0.7, 0.7, 0.7\} \{0.8, 0.7, 0.85, 0.7\} \{0.7, 0.7, 0.7, 0.8\} \{0.5, 0.5, 0.7, 0.5\}$$
$$\{0.6, 0.8, 0.85, 0.7\} \{0.7, 0.6, 0.8, 0.6\} \{0.7, 0.8, 0.7, 0.7\} \{0.8, 0.8, 0.85, 0.8\} \{0.5, 0.5, 0.5, 0.5\}$$
$$\{0.7, 0.5, 0.8, 0.5\} \{0.7, 0.7, 0.5, 0.7\} \{0.7, 0.9, 0.7, 0.8\} \{0.6, 0.6, 0.7, 0.9\})$$

The maximum length of the normalized hesitant fuzzy preference vectors HFBO and HFOW was equal to 4. Therefore, the number of elements in the normalized hesitant fuzzy preference vectors was 4, and the number of possible values in the normalized hesitant fuzzy weights was also 4.

Based on the second HFBWM model, following relationship was established:

$$Min\,\psi_R$$

$$s.t = \begin{cases} \left| \omega_B^{(l)} - \left( \omega_B^{(l)} + \omega_1^{(l)} \right) \times \left( h_{B1}^{(l)} \right) \right| \leq \psi_R \\ \left| \omega_B^{(l)} - \left( \omega_B^{(l)} + \omega_3^{(l)} \right) \times \left( h_{B3}^{(l)} \right) \right| \leq \psi_R \\ \left| \omega_B^{(l)} - \left( \omega_B^{(l)} + \omega_4^{(l)} \right) \times \left( h_{B4}^{(l)} \right) \right| \leq \psi_R \\ \left| \omega_B^{(l)} - \left( \omega_B^{(l)} + \omega_5^{(l)} \right) \times \left( h_{B5}^{(l)} \right) \right| \leq \psi_R \\ \left| \omega_1^{(1)} - \left( \omega_W^{(l)} + \omega_1^{(l)} \right) \times \left( h_{1W}^{(1)} \right) \right| \leq \psi_R \\ \left| \omega_3^{(1)} - \left( \omega_W^{(l)} + \omega_3^{(l)} \right) \times \left( h_{3W}^{(1)} \right) \right| \leq \psi_R \\ \left| \omega_4^{(1)} - \left( \omega_W^{(l)} + \omega_4^{(l)} \right) \times \left( h_{4W}^{(1)} \right) \right| \leq \psi_R \\ \left| \omega_5^{(1)} - \left( \omega_W^{(l)} + \omega_5^{(l)} \right) \times \left( h_{5W}^{(1)} \right) \right| \leq \psi_R \\ \sum_{j=1}^{10} \omega_j^{\sigma(1)} + \omega_t^{\sigma(1)} \leq 1 + \omega_t^{\sigma\,(1)} \\ \sum_{j=1}^{10} \omega_j^{\sigma(1)} + \omega_t^{\sigma(1)} \geq 1 + \omega_t^{\sigma\,(1)} \\ \omega_j \geq 0, \; j, t = 1, 2, 3, 4, 5, 6, 7, 8, 9, 10 \\ l = 1, 2, \ldots, \omega \end{cases}$$

By solving the above equation, the value of the optimal objective function was obtained as $\psi^{(l)} = 0.014$. And the normalized fuzzy weight of indicators:

$$\{0.119, 0.098, 0.072, 0.117\}\{119, 0.109, 0.115, 0, 109\}\{0.104, 0.096, 0.048, 0.098\}\{0.0723, 0.069, 0.072, 0.046\}$$
$$\{0.0723, 0.069, 0.072, 0.0466\}\{0.0595, 0.029, 0.072, 0.0556\}\{0.0723, 0.069, 0.048, 0.029\}\{0.0483, 0.069, 0.1034, 0.098\}$$
$$\{0.0483, 0.098, 0.1034, 0.098\}\{0.029, 0.024, 0.0271, 0.029\}\{0.104, 0.055, 0.0303, 0.055\}\{0.0723, 0.069, 0.0586, 0.029\}$$
$$\{0.0297, 0.069, 0.0721, 0.068\}\{0.0483, 0.069, 0.0721, 0.068\}$$

Finally, the average normalized weights were obtained as follows:

$$\omega = \{0.1015, 0.113, 0.08665, 0.064975, 0.054025, 0.092825, 0.054575, 0.079675, 0.086925, 0.027275,$$
$$0.061075, 0.057225, 0.0597, 0.06435\}$$

According to the results, indicator C2 (Competent leadership) with a weight of 0.113 and then C1 with a weight of 0.101 (Visible leadership commitment) were the most important indicators. And C10 (Procurement) was the least important indicator with a low weight of 0.027. Therefore, the order and priority of SHW indicators based on the obtained weights were shown in Table 5.

## 3.2. Results of IVHF-DEMATEL

At this stage, after collecting the IVHF-DEMATEL questionnaires from the experts, the calculation steps to determine the relationships were carried out as follows.

**3.2.1. The results of creating the initial Interval-Valued Hesitant Fuzzy IVHF matrix with a direct relationship ($\widetilde{H}$).** At this point, a matrix containing the experts' direct evaluations was obtained; linguistic terms were translated into IVF membership degrees. The interval [0,0] was used to specify the entries of H~ where one indicator does not directly affect another indicator.

**Table 5. The results of determining and prioritizing the importance and weight of SHW indicators using the HFBWM method.**

| Prioritization indicators based on HFBWM | Wj | SHW indicators |
|---|---|---|
| 1 | 0.113 | C2 |
| 2 | 0.101 | C1 |
| 3 | 0.092 | C6 |
| 4 | 0.086 | C3 |
| 5 | 0.086 | C9 |
| 6 | 0.079 | C8 |
| 7 | 0.0649 | C4 |
| 8 | 0.064 | C14 |
| 9 | 0.061 | C11 |
| 10 | 0.059 | C13 |
| 11 | 0.057 | C12 |
| 12 | 0.054 | C5 |
| 13 | 0.054 | C7 |
| 14 | 0.027 | C10 |

**3.2.2. The results of creating the group direct-relation IVHF matrix D~.** At this point, the D^ matrix was obtained, which is shown in Table 6 and contains the total of each individual evaluation in the group evaluations as individual intervals. This stage includes an example of the calculations for expert evaluations in the indicators (C2) of Competent leadership (C3) of evaluating risk management, so that the numbers and results obtained are easier to understand. IVHFEs of experts assessments of the relationship between the indicators as follows:

$$\{[0.85, 0.95]\}, \{[0.65, 0.75]\}, \{[0.85, 0.95]\}, \{[0.85, 0.95]\}, \{[0.65, 0.75]\}, \{[0.65, 0.75]\},$$
$$\{[0.85, 0.95]\}, \{[0.85, 0.95]\}, \{[0.85, 0.95]\}, \{[0.85, 0.95]\}$$

The weight of the expert assessments was assumed to be 0.05. Individual evaluations were calculated using the following equation:

$$\tilde{d}_{13} = \{[1-(1-0.85)^{0.05} \times 1-(1-0.65)^{0.05} \times 1-(1-0.85)^{0.05} \times 1-(1-0.85)^{0.05} \times 1-(1-0.65)^{0.05} \times 1-(1-0.65)^{0.05}$$
$$\times 1-(1-0.85)^{0.05} \times 1-(1-0.85)^{0.05} \times 1-(1-0.85)^{0.05} \times 1-(1-0.85)^{0.05} = \{[0.52, 0.669]\}$$

The group is direct-relation IVHF matrix D~ resulting from the above relationship is given in Table 7.

Where the maximum sum of rows was calculated as d = max{8.34,8.95,6.15,5.51,...}=8.95

By dividing each element of D~ by the maximum value of the sum of the rows, the IVHF matrix with the direct relationship of the normalized group is obtained according to Table 8. Here, only the sum of the upper bounds is sufficient, because due to the nature of the intervals, the sum of the upper limits will always be greater than the sum of the lower limits. The first row and the third column element of the matrix $\widetilde{S}$ are obtained as follows:

$$\left\{ \left[ \widetilde{S}_{13}^{L}, \widetilde{S}_{13}^{U} \right] \right\} = \left\{ \left[ \frac{0.52}{8.95}, \frac{0.669}{8.95} \right] \right\} = (0.06, 0.07)$$

Then, the IVHF matrix is divided into two separate hesitant fuzzy matrices (SL) and (SU) with the direct relationship of the normalized group, each of which consists of the lower and upper limits of the hesitant fuzzy elements, respectively (Table 9).

**Table 6. Matrix (D~).**

| SHW indicators | C1 | | C2 | | C3 | | C4 | | C5 | | C6 | | C7 | | C8 | | C9 | | C10 | | C11 | | C12 | | C13 | | C14 | |
|---|---|---|---|---|---|---|---|---|---|---|---|---|---|---|---|---|---|---|---|---|---|---|---|---|---|---|---|---|
| | L | U | L | U | L | U | L | U | L | U | L | U | L | U | L | U | L | U | L | U | L | U | L | U | L | U | L | U |
| C1 | 0.000 | 0.000 | 0.569 | 0.730 | 0.52 | 0.669 | 0.455 | 0.583 | 0.427 | 0.569 | 0.560 | 0.715 | 0.478 | 0.616 | 0.541 | 0.691 | 0.468 | 0.616 | 0.510 | 0.656 | 0.520 | 0.673 | 0.510 | 0.656 | 0.543 | 0.656 | 0.510 | 0.656 |
| C2 | 0.494 | 0.650 | 0.000 | 0.000 | 0.560 | 0.715 | 0.531 | 0.682 | 0.731 | 0.673 | 0.578 | 0.737 | 0.531 | 0.682 | 0.560 | 0.715 | 0.516 | 0.669 | 0.550 | 0.707 | 0.541 | 0.691 | 0.715 | 0.715 | 0.578 | 0.737 | 0.578 | 0.737 |
| C3 | 0.255 | 0.346 | 0.169 | 0.224 | 0.000 | 0.000 | 0.469 | 0.609 | 0.437 | 0.519 | 0.483 | 0.632 | 0.457 | 0.597 | 0.549 | 0.601 | 0.387 | 0.502 | 0.283 | 0.377 | 0.407 | 0.511 | 0.386 | 0.503 | 0.503 | 0.417 | 0.320 | 0.417 |
| C4 | 0.174 | 0.235 | 0.164 | 0.213 | 0.430 | 0.558 | 0.000 | 0.000 | 0.340 | 0.440 | 0.355 | 0.456 | 0.634 | 0.521 | 0.370 | 0.471 | 0.345 | 0.446 | 0.291 | 0.368 | 0.397 | 0.502 | 0.503 | 0.503 | 0.459 | 0.588 | 0.212 | 0.276 |
| C5 | 0.118 | 0.175 | 0.097 | 0.137 | 0.221 | 0.307 | 0.175 | 0.232 | 0.000 | 0.000 | 0.213 | 0.274 | 0.262 | 0.349 | 0.229 | 0.297 | 0.206 | 0.268 | 0.221 | 0.305 | 0.266 | 0.358 | 0.194 | 0.252 | 0.241 | 0.241 | 0.123 | 0.171 |
| C6 | 0.314 | 0.425 | 0.243 | 0.334 | 0.422 | 0.540 | 0.269 | 0.344 | 0.199 | 0.263 | 0.000 | 0.000 | 0.312 | 0.393 | 0.464 | 0.462 | 0.285 | 0.363 | 0.252 | 0.325 | 0.342 | 0.427 | 0.349 | 0.349 | 0.363 | 0.477 | 0.329 | 0.440 |
| C7 | 0.198 | 0.276 | 0.157 | 0.207 | 0.253 | 0.322 | 0.340 | 0.440 | 0.304 | 0.411 | 0.291 | 0.368 | 0.000 | 0.000 | 0.310 | 0.406 | 0.247 | 0.317 | 0.194 | 0.254 | 0.198 | 0.276 | 0.324 | 0.425 | 0.320 | 0.312 | 0.227 | 0.312 |
| C8 | 0.273 | 0.375 | 0.292 | 0.402 | 0.422 | 0.540 | 0.360 | 0.462 | 0.279 | 0.358 | 0.414 | 0.530 | 0.546 | 0.481 | 0.000 | 0.000 | 0.325 | 0.423 | 0.299 | 0.395 | 0.340 | 0.439 | 0.487 | 0.487 | 0.345 | 0.446 | 0.253 | 0.322 |
| C9 | 0.199 | 0.263 | 0.162 | 0.201 | 0.305 | 0.399 | 0.250 | 0.339 | 0.118 | 0.160 | 0.280 | 0.356 | 0.247 | 0.317 | 0.356 | 0.397 | 0.000 | 0.000 | 0.315 | 0.410 | 0.318 | 0.429 | 0.273 | 0.363 | 0.372 | 0.257 | 0.193 | 0.257 |
| C10 | 0.261 | 0.351 | 0.131 | 0.177 | 0.238 | 0.327 | 0.174 | 0.235 | 0.188 | 0.247 | 0.250 | 0.339 | 0.211 | 0.278 | 0.380 | 0.509 | 0.348 | 0.462 | 0.000 | 0.000 | 0.241 | 0.310 | 0.297 | 0.297 | 0.280 | 0.356 | 0.229 | 0.297 |
| C11 | 0.210 | 0.290 | 0.131 | 0.177 | 0.329 | 0.440 | 0.370 | 0.471 | 0.256 | 0.344 | 0.311 | 0.404 | 0.363 | 0.477 | 0.340 | 0.439 | 0.280 | 0.356 | 0.123 | 0.171 | 0.000 | 0.000 | 0.345 | 0.446 | 0.320 | 0.207 | 0.157 | 0.207 |
| C12 | 0.220 | 0.261 | 0.118 | 0.160 | 0.349 | 0.462 | 0.320 | 0.417 | 0.168 | 0.239 | 0.291 | 0.368 | 0.375 | 0.329 | 0.319 | 0.353 | 0.325 | 0.423 | 0.181 | 0.241 | 0.223 | 0.291 | 0.000 | 0.000 | 0.316 | 0.410 | 0.169 | 0.224 |
| C13 | 0.295 | 0.386 | 0.298 | 0.406 | 0.399 | 0.527 | 0.212 | 0.276 | 0.200 | 0.260 | 0.307 | 0.386 | 0.252 | 0.325 | 0.360 | 0.460 | 0.451 | 0.579 | 0.328 | 0.442 | 0.286 | 0.361 | 0.311 | 0.404 | 0.000 | 0.000 | 0.310 | 0.406 |
| C14 | 0.224 | 0.289 | 0.253 | 0.322 | 0.241 | 0.310 | 0.258 | 0.330 | 0.275 | 0.118 | 0.217 | 0.284 | 0.286 | 0.361 | 0.336 | 0.434 | 0.403 | 0.532 | 0.235 | 0.304 | 0.206 | 0.268 | 0.206 | 0.268 | 0.334 | 0.020 | 0.000 | 0.020 |

*L and U: the lower and upper limit values of IVHF- DEMATEL numbers

**Table 7. The sum of the upper limit of the rows and the maximum value.**

|  | C1 | C2 | C3 | C4 | C5 | C6 | C7 | C8 | C9 | C10 | C11 | C12 | C13 | C14 |
|---|---|---|---|---|---|---|---|---|---|---|---|---|---|---|
| Sum U | 8.34 | 8.95 | 6.16 | 5.51 | 3.25 | 5.03 | 4.24 | 5.59 | 4.08 | 4.12 | 4.38 | 4.12 | 5.53 | 3.84 |
| MAX | 8.95 | | | | | | | | | | | | | |

**3.2.3. IVHF matrix extraction results with total relationship ($\widetilde{T}$).** In this step, by using successive exponentiation of the matrix S to raise the lower and upper limit of the matrix until reaching zero matrices, and then the sum of the resulting matrices creates the lower and upper limit of the matrix $\widetilde{T}$ respectively. To multiply the hesitant fuzzy matrix, relations 10 and 12 are used. An example of multiplying the first element of the second power of the lower limit matrix SL is shown below. To raise this matrix to the desired power, each element of the first row of the SL matrix is multiplied by the corresponding element of the first column of the same matrix. The hesitant fuzzy operation is performed as follows:

$$
\left(\widetilde{s}_{11}^{L}\right)^{(2)} = \{(0 \otimes 0) \bigoplus (0.064 \otimes 0.055) \bigoplus (0.058 \otimes 0.029) \bigoplus (0.051 \otimes 0.019)
$$
$$
\bigoplus (0.048 \otimes 0.013) \bigoplus (0.063 \otimes 0.035) \bigoplus (0.053 \otimes 0.022) \bigoplus (0.06 \otimes 0.03) \bigoplus (0.052 \otimes 0.022)
$$
$$
\bigoplus (0.057 \otimes 0.029) \bigoplus (0.058 \otimes 0.023) \bigoplus (0.057 \otimes 0.025) \bigoplus (0.061 \otimes 0.033) \bigoplus (0.057 \otimes 0.025)\}
$$
$$
= \{(0.013 + 0.0014 - 0.013 \times 0.0014)\} = \mathbf{\{0.021\}}
$$

Table 10 shows an example of the calculations to exponentiate the matrix $\widetilde{s}^{L}$.

In this study, the zero matrix was obtained for both the lower limit and the upper limit of the S^ matrix by exponentiation to 6 times. The sum of these exponentiated matrices was performed using hesitant operators as follows:

$$
\widetilde{t}_{11}^{L} = (\widetilde{S}_{11}^{L})^{1} \bigoplus (\widetilde{S}_{11}^{L})^{2} \bigoplus (\widetilde{S}_{11}^{L})^{3} \bigoplus (\widetilde{S}_{11}^{L})^{4} \bigoplus (\widetilde{S}_{11}^{L})^{5} \bigoplus (\widetilde{S}_{11}^{L})^{6}
$$

In order to obtain the result from this step and after obtaining TL and TU, they were combined to reach the limit matrix $\widetilde{T}$ as shown in Tables 11 and 12.

**3.2.4. The results of calculating the sum of the rows and the columns of the $\widetilde{T}$ matrix.** Using the IVHF summation operator, the sum of the rows showing the total influence of factor i on other indicators, and the total of the influence that factor i receives from other indicators, was obtained from the columns. For a better understanding of the obtained results, an example of calculating the sum of the first row of the $\widetilde{T}$ matrix is given as follows:

$$
\tilde{r}_{1} = \{[\tilde{r}_{1}^{L}, \tilde{r}_{1}^{U}]\}
$$

$$
\{0.035 \bigoplus 0.089 \bigoplus 0.102 \bigoplus 0.091 \bigoplus 0.085 \bigoplus 0.105 \bigoplus 0.101
$$
$$
\bigoplus 0.108 \bigoplus 0.1 \bigoplus 0.092 \bigoplus 0.1 \bigoplus 0.103 \bigoplus 0.107 \bigoplus 0.091\}
$$
$$
= \{(0.718 + 0.091 - 0.718 \times 0.091) = \{[0.75, \ 0.89]
$$

The r^ and c^ vectors are given in Table 13.

**Table 8. $\tilde{S}$ matrix.**

| SHW indicators | C1 | C2 | C3 | C4 | C5 | C6 | C7 | C8 | C9 | C10 | C11 | C12 | C13 | C14 |
|---|---|---|---|---|---|---|---|---|---|---|---|---|---|---|
| C1 | 0.00 | 0.06 | 0.06 | 0.05 | 0.05 | 0.06 | 0.05 | 0.06 | 0.05 | 0.06 | 0.06 | 0.06 | 0.06 | 0.06 |
| C2 | 0.06 | 0.00 | 0.06 | 0.06 | 0.08 | 0.06 | 0.06 | 0.06 | 0.06 | 0.06 | 0.06 | 0.08 | 0.06 | 0.06 |
| C3 | 0.03 | 0.02 | 0.00 | 0.05 | 0.05 | 0.05 | 0.05 | 0.06 | 0.04 | 0.03 | 0.05 | 0.04 | 0.06 | 0.04 |
| C4 | 0.02 | 0.02 | 0.05 | 0.00 | 0.04 | 0.04 | 0.07 | 0.04 | 0.04 | 0.03 | 0.04 | 0.06 | 0.05 | 0.02 |
| C5 | 0.01 | 0.01 | 0.02 | 0.02 | 0.00 | 0.02 | 0.03 | 0.03 | 0.02 | 0.02 | 0.03 | 0.02 | 0.03 | 0.01 |
| C6 | 0.04 | 0.03 | 0.05 | 0.03 | 0.02 | 0.00 | 0.03 | 0.05 | 0.03 | 0.03 | 0.04 | 0.04 | 0.04 | 0.04 |
| C7 | 0.02 | 0.02 | 0.03 | 0.04 | 0.03 | 0.03 | 0.00 | 0.03 | 0.03 | 0.02 | 0.02 | 0.04 | 0.04 | 0.03 |
| C8 | 0.03 | 0.03 | 0.05 | 0.04 | 0.03 | 0.05 | 0.05 | 0.00 | 0.04 | 0.03 | 0.04 | 0.05 | 0.04 | 0.03 |
| C9 | 0.02 | 0.02 | 0.03 | 0.03 | 0.01 | 0.03 | 0.03 | 0.04 | 0.00 | 0.04 | 0.04 | 0.03 | 0.03 | 0.02 |
| C10 | 0.03 | 0.01 | 0.03 | 0.02 | 0.02 | 0.03 | 0.02 | 0.04 | 0.04 | 0.00 | 0.03 | 0.03 | 0.04 | 0.03 |
| C11 | 0.02 | 0.01 | 0.04 | 0.04 | 0.03 | 0.03 | 0.04 | 0.04 | 0.03 | 0.01 | 0.00 | 0.04 | 0.04 | 0.02 |
| C12 | 0.02 | 0.01 | 0.04 | 0.04 | 0.02 | 0.03 | 0.04 | 0.04 | 0.04 | 0.02 | 0.02 | 0.00 | 0.04 | 0.02 |
| C13 | 0.03 | 0.03 | 0.04 | 0.02 | 0.02 | 0.03 | 0.03 | 0.04 | 0.05 | 0.04 | 0.03 | 0.03 | 0.00 | 0.03 |
| C14 | 0.03 | 0.03 | 0.03 | 0.03 | 0.03 | 0.02 | 0.03 | 0.04 | 0.05 | 0.03 | 0.02 | 0.02 | 0.04 | 0.00 |

**Table 9. Separation of the upper and lower limits of the $\widetilde{S}$ matrix.**

| | C1 | C2 | C3 | C4 | C5 | C6 | SL C7 | C8 | C9 | C10 | C11 | C12 | C13 | C14 |
|---|---|---|---|---|---|---|---|---|---|---|---|---|---|---|
| C1 | 0.000 | 0.064 | 0.058 | 0.051 | 0.048 | 0.063 | 0.053 | 0.060 | 0.052 | 0.057 | 0.058 | 0.057 | 0.061 | 0.057 |
| C2 | 0.055 | 0.000 | 0.063 | 0.059 | 0.082 | 0.065 | 0.059 | 0.063 | 0.058 | 0.061 | 0.060 | 0.080 | 0.065 | 0.065 |
| C3 | 0.029 | 0.019 | 0.000 | 0.052 | 0.049 | 0.054 | 0.051 | 0.061 | 0.043 | 0.032 | 0.045 | 0.043 | 0.056 | 0.036 |
| C4 | 0.019 | 0.018 | 0.048 | 0.000 | 0.038 | 0.040 | 0.071 | 0.041 | 0.039 | 0.033 | 0.044 | 0.056 | 0.051 | 0.024 |
| C5 | 0.013 | 0.011 | 0.025 | 0.020 | 0.000 | 0.024 | 0.029 | 0.026 | 0.023 | 0.025 | 0.030 | 0.022 | 0.027 | 0.014 |
| C6 | 0.035 | 0.027 | 0.047 | 0.030 | 0.022 | 0.000 | 0.035 | 0.052 | 0.032 | 0.028 | 0.038 | 0.039 | 0.041 | 0.037 |
| C7 | 0.022 | 0.018 | 0.028 | 0.038 | 0.034 | 0.033 | 0.000 | 0.035 | 0.028 | 0.022 | 0.022 | 0.036 | 0.036 | 0.025 |
| C8 | 0.030 | 0.033 | 0.047 | 0.040 | 0.031 | 0.046 | 0.061 | 0.000 | 0.036 | 0.033 | 0.038 | 0.054 | 0.039 | 0.028 |
| C9 | 0.022 | 0.018 | 0.034 | 0.028 | 0.013 | 0.031 | 0.028 | 0.040 | 0.000 | 0.035 | 0.036 | 0.030 | 0.042 | 0.022 |
| C10 | 0.029 | 0.015 | 0.027 | 0.019 | 0.021 | 0.028 | 0.024 | 0.042 | 0.039 | 0.000 | 0.027 | 0.033 | 0.031 | 0.026 |
| C11 | 0.023 | 0.015 | 0.037 | 0.041 | 0.029 | 0.035 | 0.041 | 0.038 | 0.031 | 0.014 | 0.000 | 0.039 | 0.036 | 0.018 |
| C12 | 0.025 | 0.013 | 0.039 | 0.036 | 0.019 | 0.033 | 0.042 | 0.036 | 0.036 | 0.020 | 0.025 | 0.000 | 0.035 | 0.019 |
| C13 | 0.033 | 0.033 | 0.045 | 0.024 | 0.022 | 0.034 | 0.028 | 0.040 | 0.050 | 0.037 | 0.032 | 0.035 | 0.000 | 0.035 |
| C14 | 0.025 | 0.028 | 0.027 | 0.029 | 0.031 | 0.024 | 0.032 | 0.037 | 0.045 | 0.026 | 0.023 | 0.023 | 0.037 | 0.000 |

| | C1 | C2 | C3 | C4 | C5 | C6 | SU C7 | C8 | C9 | C10 | C11 | C12 | C13 | C14 |
|---|---|---|---|---|---|---|---|---|---|---|---|---|---|---|
| C1 | 0.000 | 0.082 | 0.075 | 0.065 | 0.064 | 0.080 | 0.069 | 0.077 | 0.069 | 0.073 | 0.075 | 0.073 | 0.073 | 0.073 |
| C2 | 0.073 | 0.000 | 0.080 | 0.076 | 0.075 | 0.082 | 0.076 | 0.080 | 0.075 | 0.079 | 0.077 | 0.080 | 0.082 | 0.082 |
| C3 | 0.039 | 0.025 | 0.000 | 0.068 | 0.058 | 0.071 | 0.067 | 0.067 | 0.056 | 0.042 | 0.057 | 0.056 | 0.047 | 0.047 |
| C4 | 0.026 | 0.024 | 0.062 | 0.000 | 0.049 | 0.051 | 0.058 | 0.053 | 0.050 | 0.041 | 0.056 | 0.056 | 0.066 | 0.031 |
| C5 | 0.020 | 0.015 | 0.034 | 0.026 | 0.000 | 0.031 | 0.039 | 0.033 | 0.030 | 0.034 | 0.040 | 0.028 | 0.019 | 0.019 |
| C6 | 0.047 | 0.037 | 0.060 | 0.038 | 0.029 | 0.000 | 0.044 | 0.052 | 0.041 | 0.036 | 0.048 | 0.039 | 0.053 | 0.049 |
| C7 | 0.031 | 0.023 | 0.036 | 0.049 | 0.046 | 0.041 | 0.000 | 0.045 | 0.035 | 0.028 | 0.031 | 0.047 | 0.035 | 0.035 |
| C8 | 0.042 | 0.045 | 0.060 | 0.052 | 0.040 | 0.059 | 0.054 | 0.000 | 0.047 | 0.044 | 0.049 | 0.054 | 0.050 | 0.036 |
| C9 | 0.029 | 0.022 | 0.045 | 0.038 | 0.018 | 0.040 | 0.035 | 0.044 | 0.000 | 0.046 | 0.048 | 0.041 | 0.029 | 0.029 |
| C10 | 0.039 | 0.020 | 0.037 | 0.026 | 0.028 | 0.038 | 0.031 | 0.057 | 0.052 | 0.000 | 0.035 | 0.033 | 0.040 | 0.033 |
| C11 | 0.032 | 0.020 | 0.049 | 0.053 | 0.038 | 0.045 | 0.053 | 0.049 | 0.040 | 0.019 | 0.000 | 0.050 | 0.023 | 0.023 |
| C12 | 0.029 | 0.018 | 0.052 | 0.047 | 0.027 | 0.041 | 0.037 | 0.039 | 0.047 | 0.027 | 0.033 | 0.000 | 0.046 | 0.025 |
| C13 | 0.043 | 0.045 | 0.059 | 0.031 | 0.029 | 0.043 | 0.036 | 0.051 | 0.065 | 0.049 | 0.040 | 0.045 | 0.0 | 0.045 |
| C14 | 0.032 | 0.036 | 0.035 | 0.037 | 0.013 | 0.032 | 0.040 | 0.048 | 0.059 | 0.034 | 0.030 | 0.030 | 0.002 | 0.00 |

In order to make a comparison, the cut points were obtained by obtaining the average of the elements of the two vectors. For example, the lower limit of the cut-off point for the vertical axis was calculated using the following equation and the results were obtained:

$$r_{avg}^{L} = \frac{1}{14} \otimes (0.753 \oplus 0.789 \oplus 0.652 \oplus 0.60 \oplus 0.405 \oplus 0.574 \oplus 0.494$$
$$\oplus 0.622 \oplus 0.539 \oplus 0.49 \oplus 0.525 \oplus 0.511 \oplus 0.576 \oplus 0.522)$$

$$= \frac{1}{14} \otimes (0.753 + 0.789 - 0.753 \times 0.789) \oplus 0.652 \oplus 0.60 \oplus \ldots \oplus 0.522$$

$$= \frac{1}{14} \otimes (0.753 + 0.947 - 0.753 \times 0.947) \oplus 0.60 \oplus \ldots \oplus 0.522$$

$$= \frac{1}{14} \otimes 0.999 = 1 - (1-1)^{\frac{1}{14}} = 0.59$$

Table 10. An example of calculations to exponentiate the matrix $\widetilde{S}^L$.

| | C1 | C2 | C3 | C4 | C5 | C6 | C7 $(\widetilde{S}^L)^{(2)}$ | C8 | C9 | C10 | C11 | C12 | C13 | C14 |
|---|---|---|---|---|---|---|---|---|---|---|---|---|---|---|
| C1 | **0.021** | 0.014 | 0.026 | 0.024 | 0.023 | 0.025 | 0.028 | 0.029 | 0.026 | 0.021 | 0.024 | 0.028 | 0.028 | 0.02 |
| C2 | 0.020 | 0.019 | 0.029 | 0 | 0.022 | 0.028 | 0.031 | 0.032 | 0.029 | 0.023 | 0.026 | 0.029 | 0.031 | 0.02 |
| C3 | 0.014 | 0.013 | 0.022 | 0.017 | 0 | 0.019 | 0.022 | 0.021 | 0.020 | 0.016 | 0.018 | 0.021 | 0.020 | 0.014 |
| C4 | 0.013 | 0.011 | 0.017 | 0.018 | 0.014 | 0.017 | 0.017 | 0.02 | 0.017 | 0.01 | 0.015 | 0.017 | 0.018 | 0.014 |
| C5 | 0.007 | 0.006 | 0.010 | 0.009 | 0.008 | 0.01 | 0.010 | 0.011 | 0.01 | 0.007 | 0.009 | 0.011 | 0.010 | 0.0081 |
| C6 | 0.012 | 0.011 | 0.017 | 0.016 | 0.015 | 0 | 0.019 | 0.018 | 0.017 | 0.014 | 0.015 | 0.018 | 0.018 | 0.012 |
| C7 | 0.010 | 0.009 | 0.014 | 0.012 | 0.011 | 0.013 | 0 | 0.015 | 0.014 | 0.011 | 0.013 | 0.014 | 0.014 | 0.01 |
| C8 | 0.013 | 0.011 | 0.018 | 0.017 | 0.016 | 0.018 | 0.019 | 0 | 0.018 | 0.015 | 0.017 | 0.019 | 0.020 | 0.014 |
| C9 | 0.010 | 0.009 | 0.014 | 0.013 | 0.012 | 0.014 | 0.015 | 0.015 | 0.033 | 0.011 | 0.036 | 0.015 | 0.014 | 0.01 |
| C10 | 0.009 | 0.009 | 0.014 | 0.012 | 0.011 | 0.013 | 0.015 | 0.014 | 0.013 | 0 | 0.012 | 0.014 | 0.014 | 0.01 |
| C11 | 0.010 | 0.009 | 0.014 | 0.012 | 0.012 | 0.014 | 0.016 | 0.015 | 0.014 | 0.012 | 0 | 0.015 | 0.015 | 0.011 |
| C12 | 0.010 | 0.009 | 0.014 | 0.012 | 0.012 | 0.014 | 0.014 | 0.015 | 0.013 | 0.011 | 0.013 | 0 | 0.015 | 0.01 |
| C13 | 0.012 | 0.010 | 0.016 | 0.016 | 0.014 | 0.016 | 0.018 | 0.018 | 0.015 | 0.013 | 0.015 | 0.017 | 0 | 0.012 |
| C14 | 0.01 | 0.008 | 0.014 | 0.012 | 0.011 | 0.014 | 0.015 | 0.015 | 0.013 | 0.012 | 0.013 | 0.015 | 0.014 | 0 |

Table 11. TL matrix.

| SHWindicators | C1 | C2 | C3 | C4 | C5 | C6 | C7 TL | C8 | C9 | C10 | C11 | C12 | C13 | C14 |
|---|---|---|---|---|---|---|---|---|---|---|---|---|---|---|
| C1 | 0.0357 | 0.0891 | 0.1026 | 0.0916 | 0.0857 | 0.1054 | 0.1013 | 0.1090 | 0.1000 | 0.0928 | 0.1004 | 0.1039 | 0.1077 | 0.0912 |
| C2 | 0.0891 | 0.0333 | 0.1111 | 0.1034 | 0.1192 | 0.1112 | 0.1113 | 0.1154 | 0.1097 | 0.1000 | 0.1065 | 0.1280 | 0.1157 | 0.1008 |
| C3 | 0.0536 | 0.0413 | 0.0384 | 0.0825 | 0.0768 | 0.0881 | 0.0877 | 0.0972 | 0.0760 | 0.0576 | 0.0781 | 0.0794 | 0.0916 | 0.0623 |
| C4 | 0.0413 | 0.0368 | 0.0768 | 0.0293 | 0.0619 | 0.0683 | 0.0995 | 0.0733 | 0.0692 | 0.0560 | 0.0717 | 0.0853 | 0.0814 | 0.0468 |
| C5 | 0.0257 | 0.0215 | 0.0418 | 0.0353 | 0.0147 | 0.0404 | 0.0472 | 0.0443 | 0.0406 | 0.0382 | 0.0457 | 0.0397 | 0.0449 | 0.0272 |
| C6 | 0.0544 | 0.0446 | 0.0742 | 0.0557 | 0.0463 | 0.0296 | 0.0649 | 0.0808 | 0.0610 | 0.0509 | 0.0645 | 0.0679 | 0.0698 | 0.0556 |
| C7 | 0.0380 | 0.0316 | 0.0511 | 0.0574 | 0.0518 | 0.0540 | 0.0258 | 0.0586 | 0.0489 | 0.0383 | 0.0441 | 0.0592 | 0.0591 | 0.0425 |
| C8 | 0.0537 | 0.0520 | 0.0791 | 0.0689 | 0.0550 | 0.0773 | 0.0938 | 0.0385 | 0.0700 | 0.0596 | 0.0688 | 0.0870 | 0.0733 | 0.0538 |
| C9 | 0.0407 | 0.0342 | 0.0596 | 0.0511 | 0.0351 | 0.0562 | 0.0551 | 0.0671 | 0.0461 | 0.0552 | 0.0813 | 0.0573 | 0.0677 | 0.0416 |
| C10 | 0.0456 | 0.0300 | 0.0504 | 0.0412 | 0.0402 | 0.0508 | 0.0489 | 0.0669 | 0.0629 | 0.0202 | 0.0496 | 0.0572 | 0.0559 | 0.0437 |
| C11 | 0.0413 | 0.0306 | 0.0619 | 0.0634 | 0.0493 | 0.0592 | 0.0674 | 0.0651 | 0.0571 | 0.0350 | 0.0258 | 0.0645 | 0.0624 | 0.0375 |
| C12 | 0.0417 | 0.0288 | 0.0628 | 0.0574 | 0.0393 | 0.0562 | 0.0673 | 0.0619 | 0.0612 | 0.0403 | 0.0487 | 0.0272 | 0.0609 | 0.0381 |
| C13 | 0.0533 | 0.0507 | 0.0728 | 0.0509 | 0.0472 | 0.0628 | 0.0595 | 0.0718 | 0.0804 | 0.0600 | 0.0602 | 0.0650 | 0.0333 | 0.0570 |
| C14 | 0.0429 | 0.0434 | 0.0526 | 0.0515 | 0.0513 | 0.0493 | 0.0586 | 0.0643 | 0.0709 | 0.0470 | 0.0480 | 0.0500 | 0.0635 | 0.0207 |

The cut-off points were obtained as follows:

$$\left\{ \left[ r^L_{avg}, r^U_{avg} \right] \right\} = \left\{ \left[ c^L_{avg}, c^U_{avg} \right] \right\} = \{[0.59, 0.76]\}$$

**3.2.5. Results of integration of weights obtained from HFBWM and DEMATEL IVHF methods.** In this step, the weights obtained through HFBWM and DEMATEL -IVHF were aggregated. Aggregated weight showed the internal value of an indicator and its influence on other indicators. In doing this, first, Equation 14 was used to convert the relationships

**Table 12. TU matrix.**

| | | | | | | | TU | | | | | | | |
|---|---|---|---|---|---|---|---|---|---|---|---|---|---|---|
| SHWindicators | C1 | C2 | C3 | C4 | C5 | C6 | C7 | C8 | C9 | C10 | C11 | C12 | C13 | C14 |
| C1 | 0.076 | 0.136 | 0.167 | 0.149 | 0.135 | 0.168 | 0.156 | 0.170 | 0.159 | 0.148 | 0.159 | 0.159 | 0.157 | 0.115 |
| C2 | 0.149 | 0.075 | 0.184 | 0.170 | 0.155 | 0.182 | 0.175 | 0.186 | 0.177 | 0.163 | 0.169 | 0.173 | 0.172 | 0.157 |
| C3 | 0.091 | 0.072 | 0.079 | 0.130 | 0.111 | 0.137 | 0.132 | 0.138 | 0.125 | 0.101 | 0.121 | 0.122 | 0.112 | 0.101 |
| C4 | 0.075 | 0.066 | 0.126 | 0.063 | 0.098 | 0.113 | 0.118 | 0.118 | 0.113 | 0.094 | 0.114 | 0.116 | 0.122 | 0.081 |
| C5 | 0.050 | 0.041 | 0.075 | 0.064 | 0.033 | 0.070 | 0.077 | 0.075 | 0.087 | 0.067 | 0.076 | 0.067 | 0.057 | 0.051 |
| C6 | 0.093 | 0.078 | 0.122 | 0.096 | 0.079 | 0.066 | 0.104 | 0.115 | 0.103 | 0.089 | 0.105 | 0.099 | 0.109 | 0.097 |
| C7 | 0.069 | 0.057 | 0.089 | 0.095 | 0.085 | 0.091 | 0.053 | 0.098 | 0.087 | 0.072 | 0.080 | 0.096 | 0.083 | 0.075 |
| C8 | 0.092 | 0.087 | 0.127 | 0.112 | 0.093 | 0.123 | 0.117 | 0.075 | 0.114 | 0.100 | 0.111 | 0.116 | 0.111 | 0.088 |
| C9 | 0.067 | 0.055 | 0.095 | 0.084 | 0.059 | 0.089 | 0.084 | 0.096 | 0.053 | 0.086 | 0.093 | 0.088 | 0.076 | 0.068 |
| C10 | 0.076 | 0.054 | 0.089 | 0.074 | 0.067 | 0.088 | 0.081 | 0.107 | 0.101 | 0.045 | 0.082 | 0.082 | 0.086 | 0.073 |
| C11 | 0.071 | 0.054 | 0.102 | 0.100 | 0.080 | 0.097 | 0.103 | 0.103 | 0.092 | 0.064 | 0.053 | 0.099 | 0.074 | 0.065 |
| C12 | 0.067 | 0.051 | 0.102 | 0.092 | 0.067 | 0.090 | 0.086 | 0.092 | 0.096 | 0.070 | 0.080 | 0.052 | 0.092 | 0.065 |
| C13 | 0.093 | 0.088 | 0.126 | 0.094 | 0.083 | 0.110 | 0.102 | 0.121 | 0.130 | 0.105 | 0.103 | 0.109 | 0.107 | 0.098 |
| C14 | 0.068 | 0.066 | 0.084 | 0.081 | 0.053 | 0.080 | 0.086 | 0.097 | 0.104 | 0.074 | 0.075 | 0.077 | 0.049 | 0.042 |

**Table 13. The vectors obtained from r˄ and c˄, their average and weight obtained from IVHF- DEMATEL.**

| | r | C | R Crisp | C Crisp | Wdj |
|---|---|---|---|---|---|
| C1 | {[0.753,0.892]} | {[0.494,0.695]} | 0.822 | 0.595 | 0.076 |
| C2 | {[0.789,0.918]} | {[0.444,0.639]} | 0.853 | 0.541 | 0.076 |
| C3 | {[0.652,0.811]} | {[0.625,0.811]} | 0.731 | 0.718 | 0.077 |
| C4 | {[0.608,0.776]} | {[0.589,0.774]} | 0.692 | 0.681 | 0.073 |
| C5 | {[0.405,0.601]} | {[0.56,0.716]} | 0.503 | 0.638 | 0.06 |
| C6 | {[0.574,0.759]} | {[0.62,0.797]} | 0.667 | 0.709 | 0.073 |
| C7 | {[0.494,0.692]} | {[0.651,0.791]} | 0.593 | 0.721 | 0.07 |
| C8 | {[0.622,0.787]} | {[0.662,0.816]} | 0.705 | 0.739 | 0.07 |
| C9 | {[0.539,0.68]} | {[0.636,0.805]} | 0.609 | 0.721 | 0.07 |
| C10 | {[0.496,0.684]} | {[0.545,0.74]} | 0.59 | 0.642 | 0.065 |
| C11 | {[0.525,0.701]} | {[0.612,0.778]} | 0.613 | 0.695 | 0.069 |
| C12 | {[0.511,0.683]} | {[0.645,0.786]} | 0.597 | 0.716 | 0.07 |
| C13 | {[0.576,0.788]} | {[0.654,0.775]} | 0.682 | 0.714 | 0.07 |
| C14 | {[0.522,0.659]} | {[0.527,0.708]} | 0.591 | 0.618 | 0.064 |

obtained through DEMATEL into weights. The weights obtained through HFBWM were summarized according to equation 15 in Table 14.

And finally, the integrated weights were calculated as follows:

According to the comparative analysis of the findings presented in Table 14, it is evident that the weight derived from the IVHF- DEMATEL technique differs from the weight derived from the HFBWM approach in certain indicators. To illustrate, in the case of indicator C10, the weight obtained through HFBWM stands at 0.027, whereas it increases to 0.065 in the weight obtained through IVHF- DEMATEL. Similarly, C7 exhibits a weight of 0.054 in HFBWM, which subsequently increases to 0.07 in IVHF- DEMATEL. Furthermore, the weight attributed to the C2 indicator in the HFBWM approach decreases from 0.113 to 0.074. These variations can be attributed to the influence of interrelationships between indicators.

**Table 14. The results of the combined weights of HFBWM and IVHF- DEMATEL and their prioritization.**

| Prioritization based on aggregate weight | Wfj | Wdj | Wj | SHWindicators |
|---|---|---|---|---|
| 1 | 0.118 | 0.076 | 0.113 | C2 |
| 2 | 0.107 | 0.076 | 0.101 | C1 |
| 3 | 0.094 | 0.073 | 0.092 | C6 |
| 4 | 0.092 | 0.077 | 0.086 | C3 |
| 5 | 0.085 | 0.07 | 0.086 | C9 |
| 6 | 0.084 | 0.07 | 0.079 | C8 |
| 7 | 0.065 | 0.073 | 0.064 | C4 |
| 8 | 0.061 | 0.07 | 0.059 | C13 |
| 9 | 0.059 | 0.069 | 0.061 | C11 |
| 10 | 0.057 | 0.064 | 0.064 | C14 |
| 11 | 0.055 | 0.07 | 0.057 | C12 |
| 12 | 0.053 | 0.07 | 0.054 | C7 |
| 13 | 0.045 | 0.06 | 0.054 | C5 |
| 14 | 0.025 | 0.065 | 0.027 | C10 |

Wj: Weights obtained from the HFBWM method.

Wdj: Weights obtained from the IVHF- DEMATEL method.

Wfj: Combined weights (HFBWM and IVHF- DEMATEL).

**2.3.6. The results of determining the order of rank and drawing the influence- dependence chart.** The IVHF-DEMATEL method was used to interpret results by displaying R and C values in a chart. The vertical axis shows the influence of each indicator on other indicators, while the horizontal axis shows the total column values. High values indicate high dependency on each other. The results were displayed in Fig 5, with rectangular shapes based on the values of r and c. The cut-off points are gray vertical and horizontal lines, with fuzzy distances indicating uncertain regions. Three elements

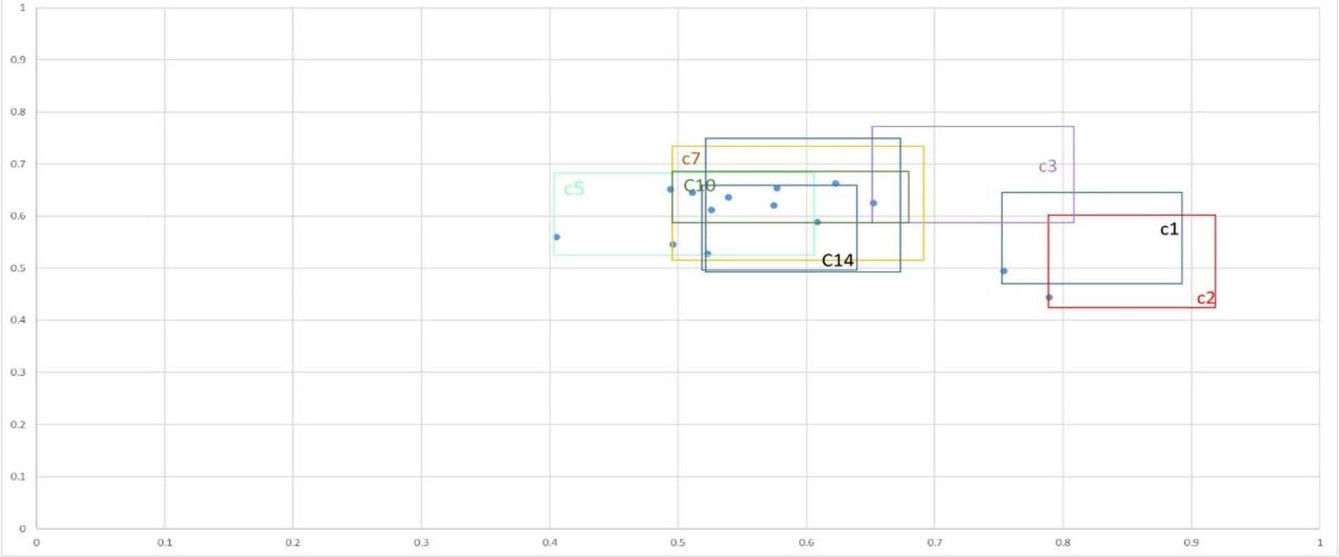

**Fig 5. The influence- dependence chart with cut-off point values (averages).**

provide a foundation for comparison: distance, relative size, and position. Distance describes variations of indicators relative to the sum of rows and columns' values. Some indicators, like C2, C1, and C5, are marginally different, while others overlap, causing ambiguity. Relative size indicates the degree of uncertainty associated with an indicator. Indicators C9, C11, C5, and C7 showed a high degree of doubt, followed by C6, C13, C2, C3, and C1. C4 and C14 showed smaller sizes, suggesting less cautious evaluation by experts. Position components place indicators in an influence-dependency chart, with rectangles representing uncertainty in their total number of rows and columns. For example, it seems that some C8, C3, and some C13 were in the critical region; however, the approach used here showed that most of C2 and some C1 were in the dependent region, which led to the conclusion that C1 was somewhat dependent and the C2 index was primarily dependent. Incorporating uncertainty in decision-making helps understand the function of an indicator in a system, enabling more accurate assessments and preventing misleading results. This analysis revealed that sensitivity analysis is necessary for the final decision, even on a qualitative level. The uncertainty region classifies indicators according to their position in the influence-dependence chart, presenting two scenarios—optimistic and pessimistic—to help decision-makers obtain new perspectives. The optimistic view considers the lower limits of the cut-off points, i.e., $c_{avg}^{L}$ and $r_{avg}^{L}$. Decision-makers can find more significant variables with the average L, but because it focuses on a smaller region, fewer factors may be found at the upper limits of the cut-off points, c_avg^L and r_avg^L. Since both dependent and critical areas are affected and change in the pessimistic scenario, there are less issues to deal with. Fig 6 shows influence-dependence chart drawn based on optimistic and pessimistic scenarios. The following results can be obtained by comparing scenarios: a) According to the optimistic scenario, the critical area comprised C8, C3, C4, most of C13 and C9, a part of C11, C14, C6, C12, C5, C10, and C7, as well as a small part of C2 and C5. On the other hand, under the pessimistic scenario, a negligible fraction of C3 and C8 were found in the critical region. (b) The optimistic viewpoint suggested that C8 and C3 indicators were influential, while the pessimistic viewpoint claimed that none of the indicators were significant in the area and that only a part of C8 and C3 belonged to the influential region. c) The pessimistic view suggests that dependent intensity indicators C14, C4, C5, C7, C11, and C10, as well as most C12, C13, C3, and C9, are excluded from the region and are neither dependent nor influential. In the optimistic scenario, most C1 and C2 are in the dependent region, while C14, C7, and a small part of C5 are related to the region and are dependent indicators.

## 3.3. Results of HF-AD

Using the HF-AD method, 16 Iran petrochemical companies were ranked at this point based on 14 proactive leading indicators for SHW. 47 experts in the field of OSH with at least five years of work experience participated in this study, with an average of two experts/specialists from each petrochemical company. Initially, the experts filled out the questionnaire (Design range) to ascertain the DR for each index. Each index's minimum point corresponded to the minimum DR value. The term "design range" refers to the range within which the system must be installed in order to achieve the desired status. Table 15 shows the DR values for this study.

The current situation (System Range) of VZ proactive leading indicators for SHW in the studied petrochemical companies was determined by experts. The results show different scores, which indicates doubt in the answers. To model uncertainty, fuzzy linguistic terms and triangular fuzzy numbers (TFN) related to indicators were used. Then, the information content of the alternatives was calculated based on the indicators. For example, the information content of C1 for petrochemical company P1 is as follows:

P1 System Range = {fair,good}

Design range P1 = {at laest good}

System Range = (80-40)/2)*1+(60-20)/2*1-(60-40/2)*0.5=35

Common range = (80-40)/2*0.5=10

Fig 7 shows the common range, design range and system range. Table 16 shows the calculation results of system range (SR) and common range (CR).

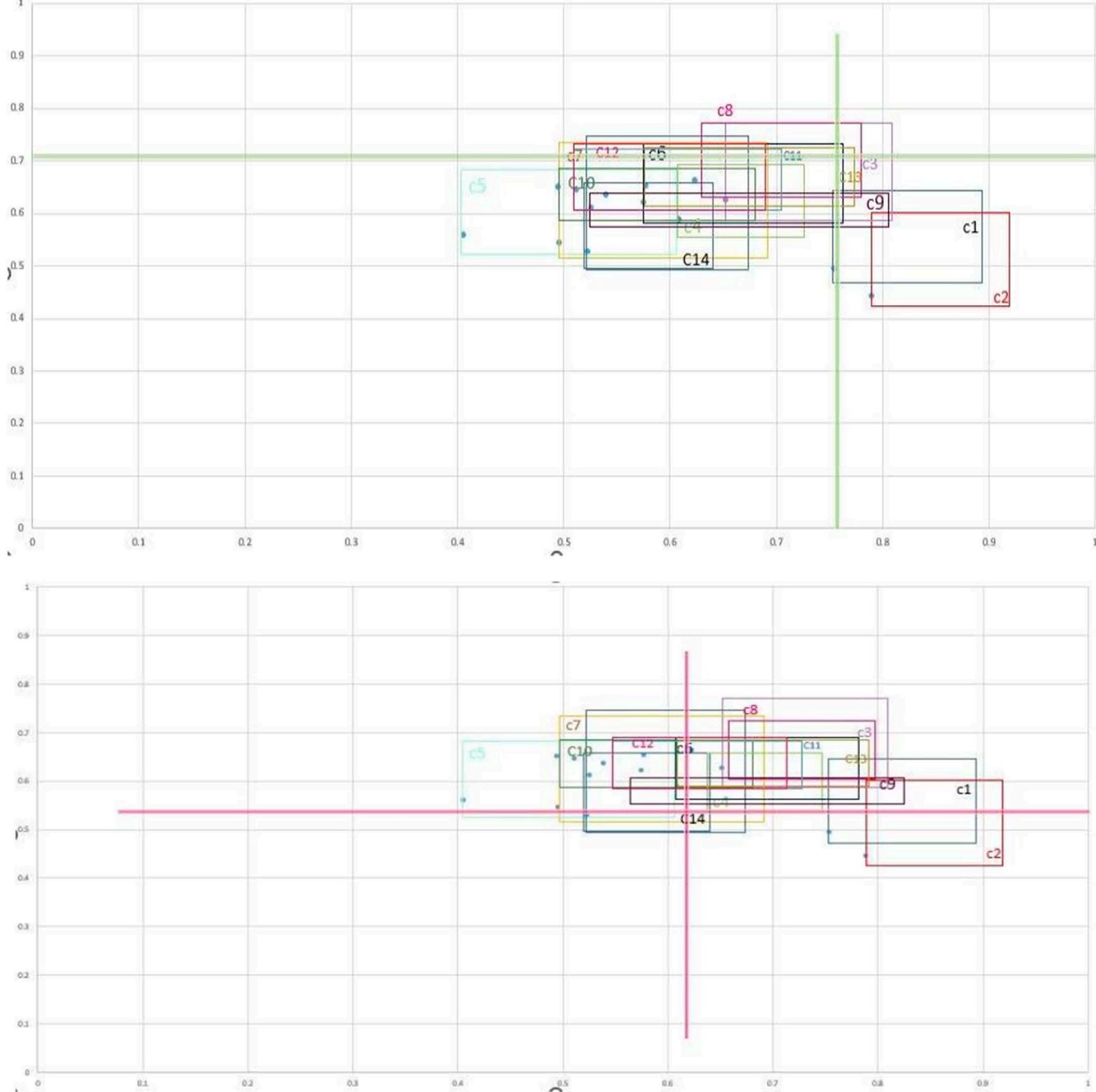

**Fig 6. The influence- dependence chart according to the optimistic scenario (a) and the pessimistic scenarios (b).**

After calculating the system range and common ranges, the information content was calculated. For example, the information content of C1 for alternative 1 is equal to:

$$IC1 = log2\ (System\ Range\ /Common\ Range) = Log\ 2\ 35/10 = 0.54$$

**Table 15. DR values.**

| Indicators | Design range |
|---|---|
| C1 | At least good |
| C2 | At least good |
| C3 | At least good |
| C4 | At least good |
| C5 | At least fair |
| C6 | At least good |
| C7 | At least good |
| C8 | At least good |
| C9 | At least good |
| C10 | At least good |
| C11 | At least good |
| C12 | At least good |
| C13 | At least good |
| C14 | At least good |

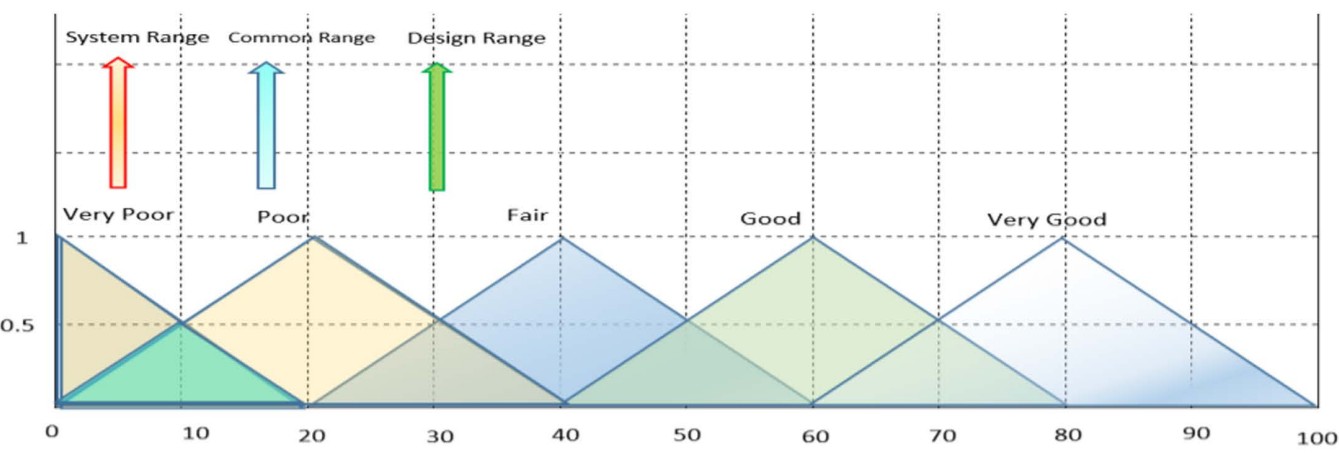

**Fig 7. Common range, design range and system range.**

Table 17 shows the rest of the information content after the calculation.

In cases where there is no common range between the system range and the design range in an alternative, the information content shall be calculated based on the infinity equation. The alternative in this case is inappropriate because it does not fit the design range of the indicated indicator. These conditions were demonstrated by the petrochemical alternatives of P5, P6, P10, P11, P13, and P14 in this study. Table 17 shows these indicators with a circle and the number 1. The number one is chosen because it has the highest information content and is therefore the maximum penalty that can be applied to an alternative that does not share a range with the design range. In Table 18, the prioritization of the studied petrochemical companies is calculated based on the combined weight SHW PLI in Table 18. As can be seen, alternative 16 (P16) with the lowest information content is the best petrochemical company, followed by alternative 8 (P8) and 9 (P9).

Table 16. Calculation results of system range (SR) and common range (CR).

| SHW indicators | C1 | | C2 | | C3 | | C4 | | C5 | | C6 | | C7 | | C8 | | C9 | | C10 | | C11 | | C12 | | C13 | | C14 | |
|---|---|---|---|---|---|---|---|---|---|---|---|---|---|---|---|---|---|---|---|---|---|---|---|---|---|---|---|---|
| | SR | CR | SR | CR | SR | CR | SR | CR | SR | CR | SR | CR | SR | CR | SR | CR | SR | CR | SR | CR | SR | CR | SR | CR | SR | CR | SR | CR |
| P1 | 35 | 10 | 50 | 20 | 50 | 20 | 35 | 20 | 35 | 20 | 35 | 20 | 35 | 20 | 20 | 20 | 65 | 10 | 50 | 20 | 35 | 5 | 50 | 20 | 50 | 20 | 40 | 10 |
| P2 | 35 | 5 | 50 | 20 | 35 | 20 | 50 | 20 | 35 | 20 | 35 | 20 | 50 | 20 | 35 | 20 | 35 | 5 | 20 | 5 | 50 | 20 | 20 | 20 | 50 | 20 | 40 | 20 |
| P3 | 50 | 20 | 65 | 20 | 55 | 20 | 55 | 20 | 65 | 20 | 50 | 20 | 50 | 20 | 35 | 5 | 35 | 20 | 50 | 20 | 55 | 20 | 55 | 20 | 55 | 20 | 50 | 20 |
| P4 | 50 | 20 | 50 | 20 | 35 | 20 | 50 | 20 | 50 | 20 | 35 | 20 | 50 | 20 | 50 | 20 | 20 | 10 | 35 | 20 | 50 | 20 | 50 | 20 | 50 | 20 | 35 | 5 |
| P5 | 40 | 5 | 40 | 5 | 30 | 0 | 35 | 20 | 35 | 20 | 35 | 5 | 50 | 20 | 35 | 20 | 35 | 5 | 35 | 5 | 20 | 20 | 35 | 20 | 55 | 20 | 45 | 20 |
| P6 | 20 | 5 | 20 | 5 | 35 | 20 | 35 | 20 | 35 | 20 | 35 | 5 | 20 | 20 | 20 | 20 | 20 | 0 | 20 | 5 | 35 | 20 | 35 | 20 | 35 | 5 | 25 | 0 |
| P7 | 35 | 20 | 35 | 20 | 35 | 5 | 35 | 20 | 40 | 20 | 35 | 20 | 35 | 20 | 40 | 10 | 40 | 20 | 40 | 20 | 35 | 20 | 35 | 20 | 40 | 10 | 35 | 20 |
| P8 | 35 | 5 | 40 | 20 | 20 | 20 | 20 | 20 | 35 | 20 | 20 | 20 | 20 | 20 | 35 | 20 | 20 | 5 | 40 | 20 | 20 | 20 | 35 | 20 | 35 | 20 | 40 | 20 |
| P9 | 20 | 5 | 20 | 5 | 20 | 5 | 20 | 20 | 20 | 20 | 20 | 20 | 20 | 20 | 20 | 20 | 20 | 5 | 20 | 20 | 20 | 20 | 20 | 5 | 20 | 5 | 20 | 5 |
| P10 | 20 | 5 | 35 | 5 | 35 | 5 | 35 | 5 | 35 | 20 | 35 | 5 | 35 | 5 | 35 | 5 | 35 | 5 | 35 | 5 | 20 | 20 | 20 | 0 | 40 | 20 | 40 | 20 |
| P11 | 20 | 0 | 20 | 5 | 20 | 0 | 35 | 5 | 20 | 5 | 20 | 0 | 35 | 5 | 20 | 0 | 35 | 5 | 20 | 0 | 20 | 5 | 35 | 5 | 20 | 0 | 25 | 0 |
| P12 | 35 | 5 | 35 | 20 | 35 | 20 | 35 | 20 | 35 | 20 | 35 | 20 | 50 | 20 | 35 | 20 | 35 | 20 | 35 | 20 | 40 | 10 | 40 | 10 | 35 | 20 | 35 | 20 |
| P13 | 50 | 10 | 50 | 20 | 50 | 20 | 50 | 20 | 35 | 5 | 50 | 20 | 40 | 0 | 35 | 20 | 50 | 20 | 55 | 20 | 65 | 10 | 35 | 20 | 55 | 20 | 40 | 10 |
| P14 | 20 | 5 | 35 | 5 | 40 | 20 | 20 | 0 | 40 | 20 | 35 | 5 | 35 | 20 | 35 | 5 | 25 | 5 | 35 | 5 | 40 | 20 | 50 | 20 | 35 | 5 | 35 | 5 |
| P15 | 20 | 20 | 35 | 5 | 35 | 5 | 35 | 20 | 35 | 5 | 35 | 5 | 35 | 5 | 35 | 5 | 20 | 20 | 20 | 20 | 20 | 20 | 35 | 5 | 20 | 20 | 35 | 5 |
| P16 | 35 | 5 | 35 | 20 | 35 | 20 | 35 | 20 | 20 | 20 | 35 | 20 | 35 | 20 | 35 | 20 | 20 | 20 | 35 | 20 | 35 | 20 | 35 | 20 | 35 | 20 | 35 | 20 |

### 3.4. The results of evaluating the current status of SHW indicators with the IAM-VZPLI

In Table 19, the results related to the frequency percentage of based on the IAM-VZPLI (second option) were presented. The color and numbers of the cells indicate the SHW indicators status based on the IAM-VZPLI. As it can be seen, none of the petrochemical companies managed to be "achieving" in all three aspects of SHW, and only P3, P12 and P15 companies managed to do so only in the safety aspect. On the other hand, none of the studied companies were at the lowest level of establishment (Starting). In safety and health aspects, 43.75% and 62.5% of the companies, respectively, were classified as "Advancing" and 37.5% of the companies were classified as "Learning" for the welfare indicators, which shows the attention It is insufficient in this area compared to the other two areas.

Fig 8 shows The results of evaluating of safety, health and wellbeing aspects in the studied petrochemical companies with IAM-VZPL.

### 3.5. Comparing the evaluation results of IAM-VZPLI with EMA-VZPLI

In Table 20, the results of the ranking of the studied petrochemical companies with two methods, IAM-VZPLI and EMA-VZPLI, are presented, and the results showed that the ranking made in these two methods was different from each other, for example, in the EMA- VZPLI, P16 was the first rank, but in the IAM-VZPLI method, this petrochemical was ranked eighth. Fig 9 shows the comparative chart of prioritization of IAM-VZPLI model and EMA-VZPLI model. The advantages of the EMA-VZPLI model in this study over the methods provided by ISSA, namely IAM-VZPLI, for measuring and scoring the current state of industries include the following: The possibility of using the criteria for scoring indicators based on fuzzy numbers, the possibility of scoring by experts in conditions of uncertainty or hesitation, scoring based on the importance and impact of each indicator by calculating the weight of indicators by HFBWM and the relationships of indicators by IVHF-DEMATEL (using the results to provide control solutions to the studied industries), and calculating the final score and designing the desired state based on the HF-AD technique through the degree of conformity of the current state and the desired state.

**Table 17. Calculation results of the information content of the alternatives.**

| LOG(SR/CR) | C1 LOG(SR/CR) | C2 LOG(SR/CR) | C3 LOG(SR/CR) | C4 LOG(SR/CR) | C5 LOG(SR/CR) | C6 LOG(SR/CR) | C7 LOG(SR/CR) | C8 LOG(SR/CR) | C9 LOG(SR/CR) | C10 LOG(SR/CR) | C11 LOG(SR/CR) | C12 LOG(SR/CR) | C13 LOG(SR/CR) | C14 LOG(SR/CR) |
|---|---|---|---|---|---|---|---|---|---|---|---|---|---|---|
| P1 | **0.54** | 0.4 | 0.4 | 0.24 | 0.24 | 0.24 | 0.24 | 0 | 0.81 | 0.4 | 0.85 | 0.4 | 0.4 | 0.6 |
| P2 | 0.85 | 0.4 | 0.24 | 0.4 | 0.24 | 0.24 | 0.4 | 0.24 | 0.85 | 0.6 | 0.4 | 0 | 0.4 | 0.3 |
| P3 | 0.4 | 0.51 | 0.44 | 0.44 | 0.51 | 0.51 | 0.4 | 0.85 | 0.24 | 0.4 | 0.44 | 0.44 | 0.44 | 0.4 |
| P4 | 0.4 | 0.4 | 0.24 | 0.4 | 0.4 | 0.4 | 0.4 | 0.4 | 0.3 | 0.24 | 0.4 | 0.4 | 0.4 | 0.85 |
| P5 | 0.9 | 0.9 | ⊖ | 0.24 | 0.24 | 0.24 | 0.4 | 0.24 | 0.85 | 0.85 | 0 | 0.24 | 0.44 | 0.35 |
| P6 | 0.6 | 0.6 | 0.24 | 0.24 | 0.24 | 0.24 | 0 | 0 | ⊖ | 0.6 | 0.24 | 0.24 | 0.85 | ⊖ |
| P7 | 0.24 | 0.24 | 0.85 | 0.24 | 0.3 | 0.3 | 0.24 | 0.6 | 0.3 | 0.3 | 0.24 | 0.24 | 0.6 | 0.24 |
| P8 | 0.85 | 0.3 | 0 | 0 | 0.24 | 0.24 | 0 | 0.24 | 0.6 | 0.3 | 0 | 0.24 | 0.24 | 0.3 |
| P9 | 0.6 | 0.6 | 0.6 | 0 | 0 | 0 | 0 | 0 | 0.6 | 0 | 0 | 0.6 | 0.6 | 0.6 |
| P10 | 0.6 | 0.85 | 0.85 | 0.85 | 0.24 | 0.24 | 0.85 | 0.85 | 0.85 | 0.85 | 0 | ⊖ | 0.3 | 0.3 |
| P11 | ⊖ | 0.6 | ⊖ | 0.85 | 0.6 | 0.6 | 0.85 | ⊖ | 0.85 | ⊖ | 0.6 | 0.85 | ⊖ | ⊖ |
| P12 | 0.85 | 0.24 | 0.24 | 0.24 | 0.24 | 0.24 | 0.4 | 0.24 | 0.24 | 0.24 | 0.6 | 0.6 | 0.24 | 0.24 |
| P13 | 0.7 | 0.4 | 0.4 | 0.4 | 0.85 | 0.85 | ⊖ | 0.24 | 0.4 | 0.44 | 0.81 | 0.24 | 0.44 | 0.6 |
| P14 | 0.6 | 0.85 | 0.3 | ⊖ | 0.3 | 0.3 | 0.24 | 0.85 | 0.7 | 0.85 | 0.3 | 0.4 | 0.85 | 0.85 |
| P15 | 0 | 0.85 | 0.85 | 0.24 | 0.85 | 0.85 | 0.85 | 0.85 | 0 | 0 | 0 | 0.85 | 0 | 0.85 |
| P16 | 0.85 | 0.24 | 0.24 | 0.24 | 0 | 0 | 0.24 | 0.24 | 0 | 0.24 | 0.24 | 0.24 | 0.24 | 0.24 |
| Wfj | 0.118 | 0.107 | 0.094 | 0.092 | 0.085 | 0.084 | 0.065 | 0.061 | 0.059 | 0.057 | 0.055 | 0.053 | 0.045 | 0.025 |

$C_1$–$C_{14}$: ISSA proactive leading indicators for SHW (Table 1).

$P_1$–$P_{16}$: Iran's Petrochemical Industries.

**Table 18. Information content and ranking of alternatives.**

| Rank | I (Information content) | Petrochemical companies |
|------|-------------------------|-------------------------|
| 1 | 0.253 | P16 |
| 2 | 0.284 | P8 |
| 3 | 0.346 | P9 |
| 4 | 0.356 | P12 |
| 5 | 0.365 | P7 |
| 6 | 0.397 | P4 |
| 7 | 0.412 | P2 |
| 8 | 0.413 | P1 |
| 9 | 0.444 | P6 |
| 10 | 0.466 | P3 |
| 11 | 0.522 | P15 |
| 12 | 0.539 | P5 |
| 13 | 0.542 | P13 |
| 14 | 0.603 | P14 |
| 15 | 0.63 | P10 |
| 16 | 0.834 | P11 |

**Table 19. The results of SHW indicators status based on the IAM-VZPLI.**

| Number of companies (percentage) | | | | |
|------|------|------|------|------|
| Safety | Health | Well-being | Status | Range of earned points (percentage) |
| 18.77% | 0% | 0% | Achieving | 81-100% |
| 43.75% | 62% | 31.25% | Advancing | 61-80% |
| 31.25% | 18.75% | 31.25% | Progressing | 41-60% |
| 6.20% | 18.75% | 37.25% | Learning | 21-40% |
| 0% | 0% | 0% | Starting | 0-20% |

## 4. Discussion

This study used the MCDM method, consists of the HFBWM, IVHF-DEMATEL, and HF-AD methods, to design a model for evaluating VZ proactive leading indicators for SHW in Iran's Petrochemical Industries. The findings showed that expert uncertainty, weighing indicators, and analyzing relationships can provide a more accurate picture of the current status of these indicators. This approach can help achieve VZ goals and achieve the desired status.

### 4.1. Comparative study of the indicators of the VZ with the requirements of other OHSMS

In a study aimed at providing tools for managing and implementing SHW, researchers examined the VZ and its theoretical aspects and the process of its application. The importance of SHW for sustainable development in the mining sector and at the national level has been recognized by mining companies. Therefore, these companies have implemented many measures to improve their safety and health management achievements, which, in addition to technological measures, have been prioritized by organizational and personal measures. Overall, this study showed that the Seven Golden Rules of VZ are gradually introduced in coal mining companies in Vietnam at different levels of training for awareness, implementation of rules and development in various activities such as the design of manuals and instructions in mine [33]. The present study showed that the implementation of OHSMS can partially achieve VZ goals in the aspects of SHW, but

it does not necessarily lead to the complete realization of these goals.This is because these systems are part of the VZ goal, and the strategy has a vision that goes beyond these systems, making the OHSMS one of the VZ goals. Therefore, this study suggests that companies should prioritize employee and manager attitudes towards SHW. Implementing ISO 45001, HSE-MS, and PSM(Like P16) can prioritize compliance with desirable conditions based on SHW indicators. However, companies that only have ISO 45001(Like P8) may be prioritized due to its compatibility with SHW indicators.

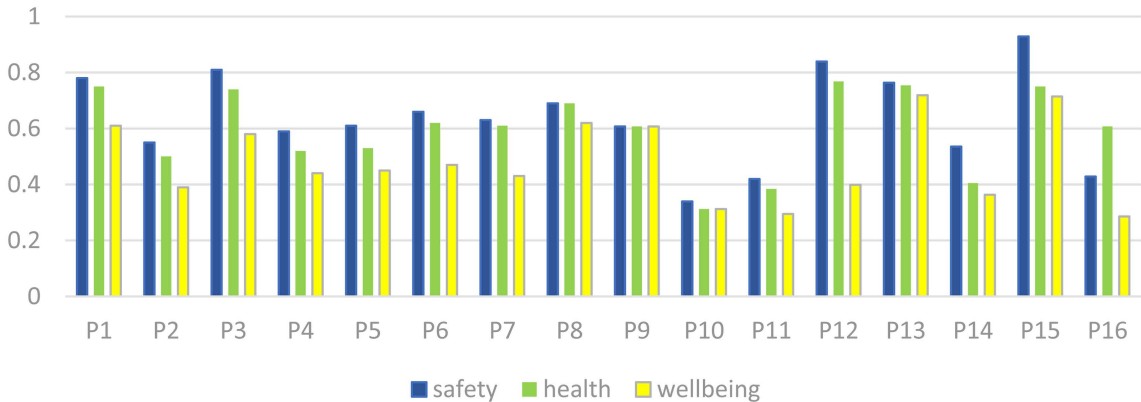

**Fig 8. The results of evaluating of safety, health and wellbeing aspects in the studied petrochemical companies with IAM-VZPL.**

**Table 20. Comparison of ranking of the companies based EMA-VZPLI and IAM-VZPLI.**

| Method IAM-VZPLI | | Method EMA-VZPLI | | Ranking |
|---|---|---|---|---|
| Petrochemical Company Code | Average score | Petrochemical Company Code | I (information content) | |
| P15 | 10.786 | P16 | 0.253 | 1 |
| P13 | 8.946 | P8 | 0.284 | 2 |
| P3 | 8.557 | P9 | 0.346 | 3 |
| P8 | 8.071 | P12 | 0.356 | 4 |
| P12 | 8.036 | P7 | 0.365 | 5 |
| P1 | 7.857 | P4 | 0.397 | 6 |
| P9 | 7.286 | P2 | 0.412 | 7 |
| P16 | 7.214 | P1 | 0.413 | 8 |
| P6 | 7.071 | P6 | 0.444 | 9 |
| P7 | 6.75 | P3 | 0.466 | 10 |
| P5 | 6.4 | P15 | 0.522 | 1 |
| P4 | 6.196 | P5 | 0.539 | 12 |
| P2 | 5.786 | P13 | 0.542 | 13 |
| P14 | 5.215 | P14 | 0.603 | 14 |
| P11 | 4.393 | P10 | 0.63 | 15 |
| P10 | 3.857 | P11 | 0.834 | 16 |

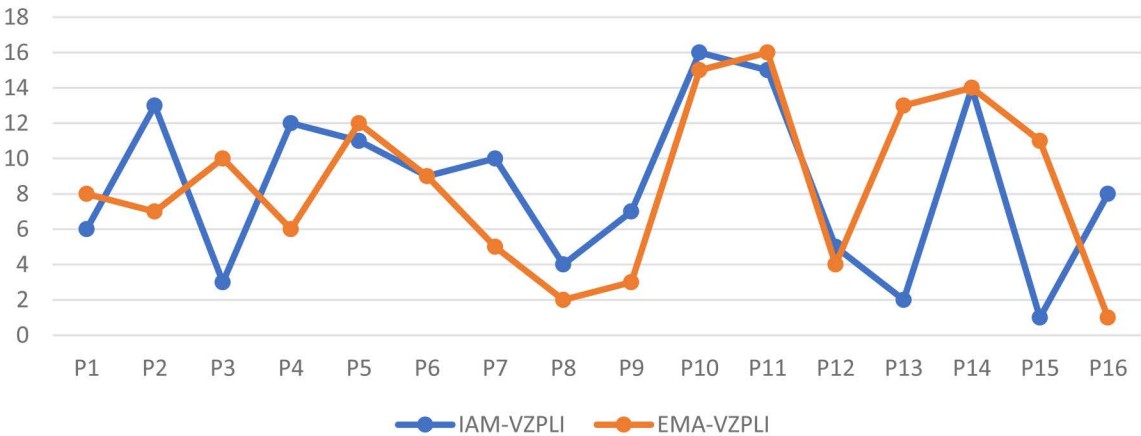

**Fig 9. Comparative chart of prioritization of IAM-VZPLI model and EMA-VZPLI model.**

#### 4.2. Assessing the importance and weight of each SHW proactive indicators

Using the HFBWM method to determine the weights and importance of the VZ Proactive leading indicators for SHW showed that the C2 indicator (competent leadership) was the most important, while the C10 indicator (Procurement) was the least important. This highlights the importance of leadership in the VZ strategy and OHSMS(Including ISO45001 and HSE-MS). The study emphasizes the need for organizations to pay attention to this aspect, as competent leadership and visible leadership commitment are crucial for improving SHW. Using the HFBWM method for weighting indicators helps companies to identify high priority indicators among 14 proactive leading indicators for SHW based on the nature of their company. Zwetsloot, G., et al in their research aimed at developing preventive indicators for SHW in the workplace introduced 14 proactive SHW indicators based on the seven golden rules of the ISSA vision zero, and explained how these indicators were prepared and compiled, and how they were measured and developed by small and large companies and other countries [9]. It should be noted that in the VZ indicators, the leadership element includes two indicators: competent leadership and visible leadership commitment, which ranked first and second respectively among the 14 indicators. However, the competent leadership element was assigned more weight than visible leadership commitment, and the results indicate the need for organizations to pay attention to this issue. According to the ISSA guidelines and report cards, in this indicator, leaders demonstrate their commitment to SHW through visible commitment and actively promote and support SHW improvement processes and the development of a prevention culture [16]. The presence of a committed and competent SHW leader is essential to advance the development processes of VZ. Such leaders demonstrate their intrinsic motivation to improve SHW and promote SHW as core personal and organizational values. Leaders then consider SHW as integrated parts of business processes and support continuous SHW improvement processes, and create a strong prevention culture [16].

#### 4.3. Determining the relationships and hierarchical structure of indicators using IVHF-DEMATEL

This study relied on the IVHF-DEMATEL method to provide a clearer understanding of the complex behavior of the VZ indicators. The results indicated that the C3 (Evaluating risk management) and C8 (Planning and organization of work) indicators were influenced to some extent. On the other hand, the C2 (competent leadership) and C1 (Visible leadership commitment) indicators were dependent, showing the highest degree of dependence. The analysis in the uncertain fuzzy environment made it possible to clearly show the doubt and uncertainty of decision-makers (DMs). According to the optimistic scenario, indicators C13 (Suggestions for improvement), C6 (Evaluating targeted programs), C9 (Innovation

and change), C11 (Initial training) and C4 (Learning from unplanned events) belonged to the critical area, and indicators C5 (Workplace and job induction), and part of C10 (Procurement), C7 (Pre-work briefings), C12 (Refresher training) were the most effective indicators. The pessimistic view showed that only a small part of C3 (Evaluating risk management) and C8 (Planning and an organization of work) belong to the critical area. And the major number of indicators, i.e., C14 (Recognition and reward), C4 (Learning from unplanned events), C5 (Workplace and job induction), C7 (Pre-work briefings), C11 (Initial training) and C10 (Procurement) and the major part of C12 (Refresher training), C13 (Suggestions for improvement), C3 (Evaluating risk management) and C9 (Innovation and change) are excluded in the region.One of the managerial consequences of the applied method is that it relies on the uncertainty and therefore encourages the decision maker not to consider his decisions as the final solution, because uncertainty should also be considered in developing indicators. In particular, this method clarifies the relationships for the decision maker and provides a clearer picture of the position and role of the factors in their mutual effects. Another important innovation of this study is the modification of weights based on the behaviors of VZ proactive leading indicators. For example, the weight of the C2 index (competent leadership), which was 0.113 in HFBWM, was reduced to 0.076. The reason for this change is the indicator has a high dependence on other indicators, which has reduced its weight. As the analysis showed, C2 belonged to the dependent area in the influence dependence (I-D) diagram. On the other hand, the weight of indicators C10 (Procurement), C4 (Learning from unplanned events), C5 (Workplace and job induction), and C7 (Pre-work briefings) increased from 0.027, 0.064, 0.054 and 0.054 to 0.065, 0.073, 0.06 and 0.07, respectively, because they were in the effective area. The reason for this increase is the high influence of these indicators.

### 4.4. Ranking of petrochemical companies through HF-AD and comparison with the IAM-VZPLI model

This study used the HF-AD methodology to compare petrochemical companies, focusing on VZ indicators for SHW. The integrated weight was calculated using the HFBWM and IVHF- DEMATEL methods, prioritizing competent leadership and provisions. The method is based on information and independence principles, making it a systematic approach to designing systems under uncertainty. The ranking of petrochemical companies can help less developed companies look up to more developed ones as role models and enhance their performance in SHW. In prioritizing the petrochemical companies concider VZ indicators in three aspects of safety, health and wellbeing Petrochemicals P15, P13, P3, P8, P12, P1, P9, P16, P6, P7, P5, P4, P2, P14, P11 and P10 respectively had a higher score and had a better situation in VZ indicators, however, in the EMA-VZPLI, studying petrochemicals P16, P8, P9, P12, P7, P4, P2, P1, P6, P3, P15, P5, P13, P14, P10 and P16 had less information content and as a result they were in better condition. As can be seen, petrochemical P16 was the eighth priority by the EMA-VZPLI. In the IAM-VZPLI, P15 is the first priority, which is the eleventh priority in the EMA-VZPLI. Also, P13 petrochemical is the second priority in the IAM-VZPLI, which is the thirteenth priority in the EMA-VZPLI. Petrochemical P9 is the third priority in the EMA-VZPLI, which is the seventh priority in the IAM-VZPLI. According to the results, considering the weight and importance of the indicators, the influence and relationships between them and the expert's doubt in evaluating the existing status can make significant changes in the results. Petrochemical companies were ranked by the EMA-VZPLI according to the information content of each indicator; P16 and P8 indicated better status. The most important indicator, C1 (visible leadership commitment), is found in the majority of companies with high information content, which means that the desired status is not currently attained. P15 and P7, however, have the least information content, indicating they match the desired status perfectly. This highlights the importance of petrochemical companies' information content in achieving their desired status. Petrochemical P11 did not achieve the desired status, but it did establish and implement the ISO 45001 OHS system. Improving these indicators can lead to improvements in VZ indicators in SHW aspects. This study addresses uncertainty in evaluating VZ proactive leading indicators for SHW by combining IVHFS with BWM, DEMATEL, and AD. It simplifies assigning membership degrees to decision problems, especially those involving groups, by capturing interpersonal and personal uncertainties. The approach streamlines decision-making by assuming only the decision maker's prior knowledge and expertise

in handling uncertainty. It allows decision makers to express membership degrees as interval values, making expert judgment simpler. Compared to traditional fuzzy sets, this approach allows modeling greater uncertainty, reducing the need for supplementary expertise [20]. The method integrates IVHFS and DEMATEL, replacing the causal diagram with an influence dependence diagram, reducing accuracy and avoiding incorrect decisions due to information loss through hesitant fuzzy operations for calculations. The EMA-VZPLI model offers advantages over IAM-VZPLI in measuring and scoring industries' current status. It allows for fuzzy number scoring criteria, expert scoring in uncertain or hesitant conditions, and scoring based on the importance and influence of each indicator. The model calculates the weight of indicators using HFBWM and relationships using IVHF- DEMATEL. The final score is calculated using HF-AD, ensuring conformity between existing and desired status. This approach can be used to provide control solutions for industries. Considering the results obtained from the two methods studied and the significant difference in the results, ISSA can combine the respondent's hesitation, the weight of the indicators, and the relationships between them in existing report cards and guides with its proposed method in order to measure the current status of companies and organizations that have joined the VZ campaign, so as to provide more accurate results of the current status and to make better and more comprehensive planning to achieve the VZ goals.

## 4.5. Actionable management recommendations

According to the study by Zwetsloot, G., et al, the strength of proactive indicators in predicting and preventing adverse outcomes lies in providing managers and organizational leaders with a complementary set of forward-looking OSH goals. Organizations also use these indicators to support communication strategies and motivate employees with the aim of transforming the organizational culture from being passive and focused on problems to a proactive and solution-oriented approach [9]. In this study, by examining the distance of the companies from the desired situation and examining the importance of the indicators, it can be seen that the indicators C1 (visible leadership commitment) and C2 (competent leadership) are of great importance and it is necessary to strengthen all the petrochemical companies studied in these elements. In order to promote and improve the competent leadership indicator and provide better performance in the petrochemical companies studied, according to the report presented by ISSA, it is recommended that leaders set standards for SHW as a role model and promote them through their behavior, verbal and non-verbal communication. They also regularly conduct patrols in the workplace and talk to workers to understand the risks of SHW at the operational level and promote SHW behavior. Leaders should ensure that SHW is an integral part of the company's formal and informal meetings and is eager to identify opportunities for improvement and is an integrated part of all business activities, including procurement, planning, human resource management, performance appraisal, incident investigation, ensuring corrective actions, follow-up and learning. They can also share SHW as core values with business partners and ensure that the company's contractors and suppliers also adhere to the organization's SHW commitments. A limitation that leaders (employers) face is that leaders cannot always be physically available and visible in all work environments and to all workers (including workers who work alone, such as truck drivers), but they must ensure that everyone is aware of their SHW commitments. To improve visible leadership commitment, selection criteria for (new) leaders should include a proven track record of consistently and effectively promoting SHW and good emotional intelligence. SHW leaders should often be involved in preventing work-related accidents or illnesses and embrace SHW as key personal values. In this regard, committed and competent leaders should understand that information about critical processes and adverse events is vital for developing SHW competencies and better performance. It should also be ensured that SHW leadership is an integral part of all leadership development or training programs. A limitation that leaders (employers) face in making tangible leadership commitments is that even competent SHW leaders may encounter ethical and practical dilemmas and unexpected and undesirable consequences, which are then viewed as opportunities for continuous learning and development.

 

## 4.6. Limitations

The study's limitations include the time-consuming and complicated calculations of the EMA-VZPLI method for industrial users. To solve this limitation, a software version is suggested as a solution. Another limitation of this study is in the EMA-VZPLI method, unlike the IAM-VZPLI method, the results were not evaluated separately in the three areas of SHW and the results were only evaluated for each of the SHW indicators. Which is suggested to be considered in future studies. By using alternative imprecise settings (such as spiral fuzzy sets and circular fuzzy sets) or alternative decision-making approaches, additional investigation can be encouraged. Our investigation employs an approach that aims to assist decision-makers instead of replacing them, which leads to our ultimate noteworthy aspect. Due to this, the principal objective of the technique is to enhance decision-makers' comprehension of the intricate composition of the system rather than clarify how the system functions.

## 5. Conclusions

The study's findings demonstrated that status VZPLIs indicators were not at the "achieving" level in all three areas of SHW. This is while the majority of the examined companies had fully implemented OHSMS and there is a lot of overlap between the requirements of OHSMS and VZPLI. This demonstrates that the objectives of the vision zero strategy go beyond those of OHSMS. On the other hand, in all studied petrochemical companies, the indicators of the "well-being" aspect had a worse condition compared to "safety" and "health" aspects, which indicates the need for companies to pay more attention to the improvement of this important aspect. Moreover, the results showed that there are significant differences in ranking of petrochemical companies for VZPLIs in EMA-VZPLI compared to IAM-VZPLI. Therefore, this method could be applied to more accurate assessment of VZPLI of vision zero goals.

## Supporting information

**S1 Data. The Excel file titled** *"ISSA edited"* **contains data related to the evaluation of safety, health, and well-being indicators in the companies studied and the assessment of the current and desired status in 5 Excel sheets.**
(XLSX)

**S2 Data. The Word file titled** *"questionnaire"* **is related to the Vision Zero Proactive Leading Indicators Questionnaire.**
(XLSX)

**S3 Data. The Excel file titled "**BWM hesitant combination 2**" contains data on indicator weights and combined weight calculations in 12 sheets.**
(XLSX)

**S1 File. google form.**
(DOCX)

## Acknowledgments

The authors are extremely grateful to all participants in the study and the companies studied.

## Author contributions

**Conceptualization:** Mehdi Jahangiri, Mina Bargar.

**Data curation:** Moslem Alimohammadlou.

**Investigation:** Mehdi Jahangiri, Mina Bargar.

**Methodology:** Mina Bargar.

**Supervision:** Mehdi Jahangiri, Moslem Alimohammadlou.

**Validation:** Mina Bargar.

**Writing – original draft:** Mehdi Jahangiri, Mina Bargar.

**Writing – review & editing:** Moslem Alimohammadlou, Sanaz Karimpour, Mojtaba Kamalinia.

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
