## [Decision Letter · Decision Letter 0]

26 Jan 2025

Dear Dr. Jahangiri,

Thank you for submitting your manuscript to PLOS ONE. After careful consideration, we feel that it has merit but does not fully meet PLOS ONE’s publication criteria as it currently stands. Therefore, we invite you to submit a revised version of the manuscript that addresses the points raised during the review process.

Please carefully read the comments provided by the following three reviewers as shown below, which are against the quality of your paper. Please revise your paper and prepare responses according to those comments.

We look forward to receiving your revised manuscript.

Kind regards,

Ta-Chung Chu

Academic Editor

PLOS ONE

“This study is part of the master's thesis of Ms. Mina Bargar, a master's student at Shiraz University of Medical Sciences. It was financially supported by the Research and Technology Deputy of Shiraz University of Medical Sciences under grant number [Grant Number: 23812]. The authors are extremely grateful to Research and Technology Deputy of Shiraz University of Medical Sciences for funding this research, without your support, this work would not have been possible.”

“This study is part of the master's thesis of Ms. Mina Bargar, a master's student at Shiraz University of Medical Sciences. It was financially supported by the Research and Technology Deputy of Shiraz University of Medical Sciences under grant number [Grant Number: 23812]. The authors are extremely grateful to Research and Technology Deputy of Shiraz University of Medical Sciences for funding this research, without your support, this work would not have been possible.”

“This study is part of the master's thesis of Ms. Mina Bargar, a master's student at Shiraz University of Medical Sciences. It was financially supported by the Research and Technology Deputy of Shiraz University of Medical Sciences under grant number [Grant Number: 23812]. The authors are extremely grateful to Research and Technology Deputy of Shiraz University of Medical Sciences for funding this research, without your support, this work would not have been possible.”

6. Please include a separate caption for each figure in your manuscript.

Reviewers' comments:

Reviewer's Responses to Questions

**Comments to the Author**

1. Is the manuscript technically sound, and do the data support the conclusions?

Reviewer #1: Partly

Reviewer #2: Yes

Reviewer #3: Yes

2. Has the statistical analysis been performed appropriately and rigorously?

Reviewer #1: Yes

Reviewer #2: Yes

Reviewer #3: Yes

3. Have the authors made all data underlying the findings in their manuscript fully available?

Reviewer #1: No

Reviewer #2: Yes

Reviewer #3: Yes

4. Is the manuscript presented in an intelligible fashion and written in standard English?

Reviewer #1: Yes

Reviewer #2: Yes

Reviewer #3: Yes

Reviewer #1: • The use of abbreviations in the title disrupts the flow, and neglecting writing conventions may create a negative first impression for the reader.

• Abbreviation usage in the abstract should follow proper conventions. When a term is mentioned for the first time, its full form should be written out, followed by its abbreviation in parentheses. Subsequently, the abbreviation alone should be used consistently throughout the text.

• Providing a list of abbreviations and their meanings immediately after the abstract can help readers follow the text more easily. This is particularly beneficial in studies where abbreviations are frequently used.

• Some references cited in the text are missing from the bibliography. For instance, 'Xiaomi Mi et al. (2019)' under section 1.1 is not included in the reference list. This inconsistency can undermine the academic reliability of the study. All references cited in the text should be cross-checked with the bibliography, and any missing sources should be added.

• The study's literature review should be strengthened. Incorporating more diverse and up-to-date references related to the topic will enhance the academic foundation of the work and better highlight its contribution to the field.

• The flowchart explaining the methodology can be included within the main text to enhance the narrative's coherence. Additionally, the lengthy calculations in sections 3.2.3 and 3.2.4 could be moved to the appendices to improve readability and prevent the main sections from becoming overly detailed.

• A sensitivity analysis based on the weights of SHW (Safety, Health, and Wellbeing) indicators should be included in the study. This will evaluate the impact of different weighting scenarios on the results, enhancing the robustness of the model and the reliability of the decision-making process.

• The text formatting, including font and font size, should be standardized throughout the document. Ensuring consistency across all sections and tables will improve the professional appearance of the study. Additionally, Table 15 should be revised.

Reviewer #2: Overall, the study presents a potentially valuable contribution to the field. However, the manuscript requires significant revisions to enhance clarity, coherence, and the overall presentation of the research. Addressing these comments will strengthen the study and improve its readability and impact.

1. Title

The title is overly complex and lacks clarity in highlighting the main focus of the study. It is recommended to simplify the title to clearly emphasize the core objective.

2. Introduction

The literature review on ISSA Proactive Leading Indicators (PLIs) in the Introduction section focuses primarily on the importance of these indicators but lacks a detailed discussion of the methods used by other scholars to analyze them. Specifically, the review should address:

(1)The methods employed by previous studies to derive these indicators.

(2)Limitations or gaps in these methods, such as areas that need improvement or issues that have been overlooked.

(3)This will provide a more comprehensive background and justify the need for the current study.

3. Innovation

The novelty of the study is not clearly articulated. The authors should explicitly state the innovative aspects of their work by comparing it with existing literature. This can be achieved by:

(1)Clearly identifying the gaps in previous studies.

(2)Highlighting how the current study addresses these gaps through its unique approach or methodology.

4. Methodology

While the study introduces the methods used, it lacks a detailed discussion on how these methods have been applied in previous studies related to ISSA PLIs. The authors should:

(1)Provide a review of relevant literature where these methods (e.g., HFBWM, IVHF-DEMATEL, HF-AD) have been used.

(2)Discuss how these methods address or improve upon the limitations of previous approaches in analyzing ISSA PLIs.

5. Calculation Process

The inclusion of detailed formulas and calculations in the main text makes the article overly complex. It is recommended to:

(1)Number the formulas for easy reference.

(2)Move the detailed calculations with specific values to the appendix.

(3)Keep the main text concise by providing only the essential expressions and focusing on the interpretation of results.

6. Comparison of IAM-VZPLI and EMA-VZPLI

Section 3.5 provides a comparison between IAM-VZPLI and EMA-VZPLI but does not offer a clear analysis of which method is superior. The authors should:

(1)Conduct a thorough analysis of the strengths and weaknesses of each method.

(2)Provide specific criteria or metrics to evaluate which method is more effective in assessing the Vision Zero Proactive Leading Indicators.

7. Discussion

The Discussion section should be more structured and go beyond merely presenting the results of the factor analysis. The authors should:

(1)Break down the discussion into clear sub-sections.

(2)Focus on how the results can be applied in practice.

(3)Provide actionable management recommendations based on the findings.

8. Consistency in Abbreviations

There is inconsistency in the use of abbreviations, such as IAM-VZPLI and IAM-VZPL (e.g., the title of Section 3.5). The authors should:

Ensure consistent use of abbreviations throughout the manuscript.

Double-check for any typographical errors to maintain clarity and coherence.

Reviewer #3: The paper demonstrates a certain level of innovation, but there are still some deficiencies that need to be improved

1. In the introduction, the author devotes a considerable amount of space to introducing the background of evaluation indicators. However, there are deficiencies in the literature review and introduction regarding the rationale for using the method proposed by the author, especially in terms of why existing methods fail to consider hesitancy and uncertainty, which requires further supplementation.

2. It is suggested that the author add preliminary knowledge in the second part, before the methodology section following the introduction. This should primarily include an introduction to existing methods, followed by a distillation of their deficiencies. Additionally, brief introductions to concepts used in the paper, such as BMW (if it refers to a specific concept or method) and hesitant fuzzy sets, should be provided to facilitate readers' comprehension.

3. The new evaluation method proposed in the paper utilizes hesitant fuzzy sets. In fact, there are currently more comprehensive fuzzy sets built upon hesitant fuzzy sets, such as probabilistic hesitant fuzzy sets and probabilistic dual hesitant fuzzy sets. The author does not provide an explanation or review of the necessity of using hesitant fuzzy sets in the introduction.

4. In the discussion section, the author does not present comparisons with other methods or robustness analysis. The discussion section does not currently demonstrate the advantages of the evaluation method proposed by the author. The writing of the discussion section resembles more of a conclusion section.

**Do you want your identity to be public for this peer review?** For information about this choice, including consent withdrawal, please see our Privacy Policy

Reviewer #1: No

Reviewer #2: No

Reviewer #3: No

---

## [Author Response · Author response to Decision Letter 1]

1 Mar 2025

Dear Dr. Ta-Chung Chu,

Thank you for reviewing our manuscript submitted to your journal. The corrections requested by the esteemed reviewers are as follows:

Reviewer #1:

• According to the respected reviewer, the abbreviations in the title were removed.

• Abbreviations used in the abstract were revised.

• A list of abbreviations is provided in a table immediately after the abstract.

• All references cited in the text were reviewed and are available in the bibliography.

• The flowchart explaining the methodology was included within the main text to enhance the narrative coherence.

• Sensitivity analysis based on the weight of SHW indicators is included in the study in Section 2.3.6.

• Text formatting, including font and font size, was reviewed and edited throughout the document.

• Table 15 was edited.

Reviewer #2:

• The title of the study has been shortened and simplified to focus on the core objective of the study.

• Consistency in abbreviations was checked throughout the manuscript and typographical errors were corrected.

• According to the opinion of the respected reviewer, the introduction was reviewed and edited, and the requested items were added. An effort has been made to articulate the innovative aspects of this research separately.

• Formulas are numbered in the text.

• Detailed and lengthy calculations have been moved to the appendix.

• Thank you for the attention and care of the respected reviewer number 2, regarding the study methodology, which the respected reviewer has pointed out, no study has used multi-criteria decision-making methods to determine the importance, weight, and relationships between these indicators. For this reason, no information has been provided on how to use these methods in the field of the study. However, an effort has been made to clarify this at the end of the introduction, and relevant items have been added to the article text.

• In section 3.5, a comparison between IAM-VZPLI and EMA-VZPLI was provided and specific criteria were provided to evaluate which method is more effective in assessing the VZPLI.

• The discussion Broke down into clear sub-sections.

• Actionable management recommendations were provided at the end of the discussion.

Reviewer #3:

• The introduction section was edited based on the reviewer's comments and the requested items were added.

• According to the respected reviewer, the advantages of the evaluation method presented in the study were added to the introduction section.

• Comparison with other methods and studies was added to discussion section.

• The discussion Broke down into clear sub-sections.

There were similar comments from some reviewers, which were corrected and added.

Kind regards,

Jahangiri

---

## [Decision Letter · Decision Letter 1]

24 Mar 2025

Evaluation of ISSA Proactive Leading Indicators for Safety, Health and Well-being; Application of  Multi-Criteria Decision-Making Methods based on Hesitant Fuzzy

PONE-D-24-40912R1

Dear Dr. Jahangiri,

We’re pleased to inform you that your manuscript has been judged scientifically suitable for publication and will be formally accepted for publication once it meets all outstanding technical requirements.

Kind regards,

Ta-Chung Chu

Academic Editor

PLOS ONE
---

## [Editor Report · Acceptance letter]

PONE-D-24-40912R1

PLOS ONE

Dear Dr. Jahangiri,

I'm pleased to inform you that your manuscript has been deemed suitable for publication in PLOS ONE. Congratulations! Your manuscript is now being handed over to our production team.

Kind regards,

on behalf of

Dr. Ta-Chung Chu

Academic Editor

PLOS ONE